# Conservation agriculture improves soil health and sustains crop yields after long-term warming

Jialing Teng[1,7], Ruixing Hou[2,7], Jennifer A. J. Dungait [3,4], Guiyao Zhou [5], Yakov Kuzyakov[6], Jingbo Zhang[1], Jing Tian [1] ✉, Zhenling Cui [1] ✉, Fusuo Zhang [1] & Manuel Delgado-Baquerizo [5] ✉

Climate warming threatens global food security by exacerbating pressures on degraded soils under intensive crop production. Conservation agriculture is promoted as a sustainable solution that improves soil health and sustains crop yields in a changing climate, but these benefits may be affected by long-term warming. Here, we investigate the effects of conservation agriculture compared to conventional agriculture on 17 soil properties, microbial diversity and crop yields, during eight-years' experimental warming. An overall positive effect of warming on soil health over time under conservation agriculture is characterized by linear increases in soil organic carbon and microbial biomass carbon. Warming-triggered shifts in microbial biomass carbon and fungal diversity (saprogen richness) are directly linked to a 9.3% increase in wheat yields over eight years, but only under conservation agriculture. Overall, conservation agriculture results in an average 21% increase in soil health and supports similar levels of crop production after long-term warming compared to conventional agriculture. Our work provides insights into the potential benefits of conservation agriculture for long-term sustainable food production because improved soil health improves resilience to the effects of climate warming.

Global food security is challenged by climate warming and growth of the human population[1,2]. Soil degradation associated with intensive agriculture has reduced the availability of land for food production[3,4]. The far-reaching consequences of climate warming combined with the urgent demand to increase food production require climate-resilient land management options to simultaneously increase productivity and promote adaptations to climate change and its mitigation[3,5,6]. A sustainable management strategy described as 'conservation agriculture' is widely promoted as a nature-based solution to maintain food production and simultaneously promote soil health[2,7–10]. Soil health is a holistic concept that integrates the biological, physical and chemical aspects of soil, demonstrating the continued capacity of soil as a vital

[1]State Key Laboratory of Nutrient Use and Management, College of Resources and Environmental Sciences, National Academy of Agriculture Green Development, China Agricultural University, 100193 Beijing, PR China. [2]Key Laboratory of Ecosystem Network Observation and Modeling, Institute of Geographic Sciences and Natural Resources Research, Chinese Academy of Sciences (CAS), 100101 Beijing, PR China. [3]Geography, College of Life and Environmental Sciences, University of Exeter, Rennes Drive, Exeter EX4 4RJ, UK. [4]Carbon Management Centre, SRUC-Scotland's Rural College, Edinburgh EH9 3JG, UK. [5]Laboratorio de Biodiversidad y Funcionamiento Ecosistémico. Instituto de Recursos Naturales y Agrobiología de Sevilla (IRNAS), Consejo Superior de Investigaciones Científicas (CSIC), 41012 Sevilla, Spain. [6]Department of Soil Science of Temperate Ecosystems, University of Göttingen, 37077 Göttingen, Germany. [7]These authors contributed equally: Jialing Teng, Ruixing Hou. ✉e-mail: tianj@igsnrr.ac.cn; cuizl@cau.edu.cn; m.delgadobaquerizo@gmail.com

living ecosystem[3,10]. Conservation agriculture encompasses reduced or zero/no tillage, permanent soil cover, and diverse crop rotations, and is applicable in many different farming contexts[2,11–13]. A wide range of environmental benefits associated with its implementation, including improved soil health related to increased soil organic carbon (SOC) stocks and soil biodiversity[2,7,8], has led to adoption of conservation agriculture across 12.5% of arable land in one-third of countries worldwide[11]. However, a paucity of systematic and quantitative assessments of long-term climate warming effects on the potential of conservation agriculture to support soil health and crop production creates uncertainty in its efficacy under different future climate projections[13,14]. Long-term field experiments specifically comparing conservation agriculture with conventional management under warming conditions are particularly rare[12,15–17], but are urgently needed to explore the interactive effects of management and warming on crop yield and soil health[12,15,16].

Predicting the effectiveness of conservation agriculture as the climate warms is challenging because of the complex interactive effects of warming and soil management on crop yields and individual soil properties[18,19]. In general, climate warming profoundly affects agricultural activities in many ways, ranging from yield reductions to loss of SOC and ecosystem functions[2,5,20,21]. Reductions in crop yields in response to rising temperature are widely reported[22–26], especially at low latitudes[26,27], mainly due to drought, disturbed crop growth cycles, and increased pathogen pressures[25,28]. SOC is a 'master' soil health indicator that supports multiple soil functions including nutrient and water cycling and retention, soil structure formation, and ecosystem productivity[29–31]. Climate warming is anticipated to elevate global SOC losses by accelerating microbial decomposition[32–35]. In addition, rising temperatures may enhance microbial nitrogen mineralization and lead to nitrogen loss from terrestrial ecosystems[36]. In contrast, crop residue retention in conservation agriculture promotes SOC accrual directly by increasing plant biomass inputs, which also alleviates water and nutrient limitation for crops by improving soil health, and indirectly by accelerating microbial turnover and necromass accrual through organic matter supply[17,33]. The soil conditions created by conservation agriculture can counteract the negative effects of climate change on food production in some regions[37–39]. The complex interactive effects of warming and soil management create many uncertainties in prediction aggravated by the limitations of process-based crop simulation models to estimate the effects of climate change on agroecosystem functions. This is exacerbated by the urgent requirement for empirical data from long-term field warming experiments to improve models that forecast future farming conditions[40]; to date, short-term field-warming studies have largely focused on natural ecosystems[18].

Soil microbiome are major contributors to sustainable agriculture because they drive key processes in agroecosystems prerequisite to optimizing soil health and crop productivity[41]. Conservation agriculture generally promotes the size, diversity, activity and beneficial functions of soil microorganisms that contribute to soil health, including SOC accrual in available and stable pools, and crop productivity[17,42]. In particular, reduced tillage supports the development of hyphal networks and a more diverse and abundant soil fungal biomass, which contributes to multiple soil ecosystem functions including improved substrate and water supply[43–45]. However, the effects of climate warming on the observed benefits of conservation agriculture for the soil microbiome are difficult to anticipate. Intenser environmental filtering in warmed forest and grassland soils can create negative effects on fungal and bacterial diversity[46,47]. Stronger nutrient acquisition by roots induced by warming may lead to reductions in mycorrhizal dependence and arbuscular mycorrhizal fungal diversity[48]. Potential increases of soil-borne plant pathogens under warming conditions are of especial concern in crop production systems[28,49], though the increase in soil health that supports healthy crop growth may help counteract disease threats[50]. Overall, great

uncertainty remains about the response of soil microbiome to interactions between warming and crop management practices, and the effects on soil health and crop yields.

Our study aims to assess warming effects on: (i) soil health and crop yields under conservation agriculture (permanent crop residue cover and no tillage) versus conventional agriculture (crop residue removal and annual tillage); and (ii) the contribution of the soil microbiome to supporting soil health and crop yields. We hypothesize that, under warming conditions, (i) conservation agriculture improves crop productivity by increasing carbon inputs aboveground (crop residues) and belowground (crop roots) that stimulates microbial growth, promoting soil health characterized by carbon accrual; (ii) improvements in crop yield and soil health relies on changes in the soil microbiome; (iii) the benefits of conservation agriculture are cumulative over time. To test these hypotheses, we conduct an eight-year-long field experiment to investigate the effects of soil management systems (conservation agriculture versus conventional agriculture) and warming levels (warming versus ambient) on crop yields and soil health, and additionally the influence of the soil microbiome, on the North China Plain. Our experiment is conducted using a typical crop rotation system (winter wheat (*Triticum aestivum* L.)-summer maize (*Zea mays* L.)). Two levels of warming are imposed using infrared heaters: 'ambient' and '+2 °C' according to soil warming predicted by IPCC greenhouse gas scenarios rates for northern China[51]. We simultaneously assess 17 soil health indicators, microbial diversity and crop yields to demonstrate the combined effects of warming and management. We find that long-term warming increases soil health indicators and crop productivity, but only under conservation agriculture. The beneficial effects of conservation agriculture on microbial biomass carbon and SOC increase through time under long-term warming. Improved soil health and shifts in soil fungal diversity are related to the positive warming effects on crop yields over time under conservation agriculture. Our work demonstrates the potential for climate resilient farming through the implementation of conservation agriculture to improve soil health and sustain crop yields in a warming climate.

## Results

### Similar wheat yields in conservation agriculture and conventional agriculture under warming

Experimental warming was imposed using infrared heaters to plots used for wheat and maize cultivation adopting conservation agriculture or conventional agriculture (Fig. 1a). Over the eight-year study period, soil temperature and moisture were recorded continuously by in-situ sensors. As expected, experimental warming increased soil temperature but decreased soil moisture under both conservation and conventional agriculture ($P < 0.05$; Fig. 1b, c; Supplementary Table 1). The warming effects were modified by soil management, and soils under conservation agriculture were cooler (14.0 °C vs. 14.7 °C) and wetter (17.1% vs. 15.9%) than under conventional agriculture ($P < 0.001$; Fig. 1b, c; Supplementary Table 1).

Winter wheat yields were strongly affected by warming ($P < 0.001$; Supplementary Table 1), but there was no difference between the two management types (Fig. 1d). Conservation agriculture maintained similar wheat yields to conventional agriculture both under ambient and warming conditions (Fig. 1d). Across the eight-year study, warming increased wheat yields by 9.3% and 11.2% under conservation and conventional agriculture, respectively, when compared with the no warming treatments ($P < 0.001$; Fig. 1d; Supplementary Table 1). We further assessed the warming effect on wheat yields over eight years using Cohen's $d$ index (Fig. 1e). Warming-induced positive effects on wheat yields strengthened with time under conservation agriculture ($P < 0.05$), but not under conventional agriculture (Fig. 1e). In contrast, the maize yields were only affected by soil management ($P < 0.001$; Supplementary Table 1), but not influenced by warming (Supplementary Fig. 1). Conservation agriculture supported larger maize yields

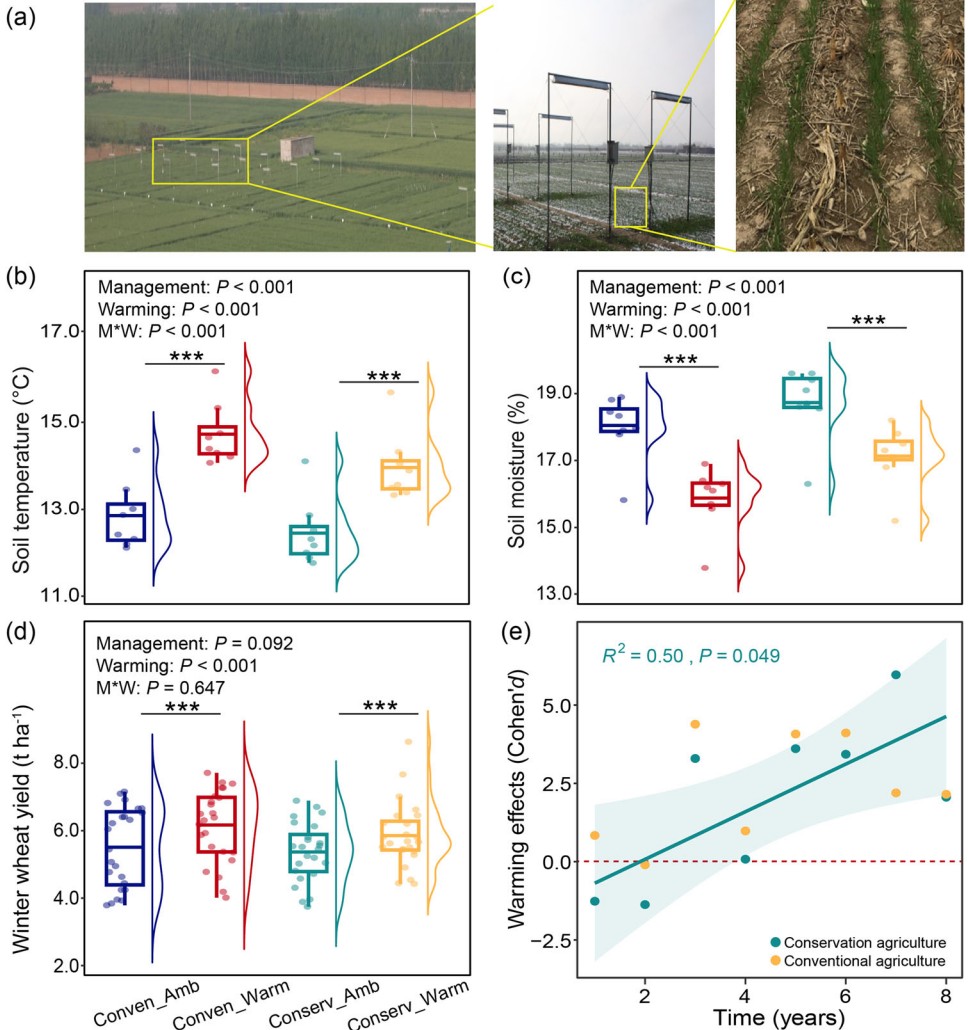

**Fig. 1 | Effects of warming and management on soil microclimate and crop yields. a** Experimental settings for treatments. **b**, **c** Effects of warming and management on soil temperature and moisture over eight years was estimated by linear mixed model with sampling time as random factors. Statistical significance is based on Wald type II χ² tests (*n* = 8 independent soil samples per treatment). All reported *P* values result from two-sided statistical tests. Boxplots display the mean (horizontal line), the 25th and 75th percentiles (colored box), the minimum and maximum (whiskers). **d** Average winter wheat yields. Boxplots display the mean (horizontal line), the 25th and 75th percentiles (colored box), the minimum and maximum (whiskers). The winter wheat yield data were analyzed based on eight sampling years (*n* = 24 independent soil samples per treatment). Statistical analysis was performed using a linear mixed model with sampling time as random factors. Statistical significance is based on Wald type II χ² tests. All reported *P* values result from two-sided statistical tests. Asterisks indicate significant differences in the warming effect of the individual management system as compared with their matched ambient condition. **e** Shift in the effect size of warming on crop yields over time for conservation and conventional agriculture, respectively. A linear regression model with two-sided test was used for the statistical analysis, and adjusted R-squared was used. Relationships are denoted with solid lines and fit statistics (R² and *P* values) for each management practice. The solid line represents the significant linear regression (*P* < 0.05), and the gray shading indicates the 95% confidence intervals. All reported *P* values result from two-sided statistical tests where **P* < 0.05, ***P* < 0.01, and ****P* < 0.001. Conserv-Amb, conservation agriculture without warming; Conserv-Warm, conservation agriculture with warming; Conven-Amb, conventional agriculture without warming; Conven-Warm, conventional agriculture with warming. Source data are provided as a Source Data file.

than conventional agriculture under both ambient and warming conditions (*P* < 0.001; Supplementary Fig. 1; Supplementary Table 1).

**Conservation agriculture improved soil health under warming**
Seventeen soil properties, including physical, chemical, and biological attributes, were analyzed in 2020 using samples from two soil depths (0–5 cm and 5–15 cm depth) to evaluate the cumulative effects of long-term warming on soil health. Principal Component Analysis (PCA) was used to select representative indicators of soil health (Fig. 2a; Supplementary Fig. 2). The first two principal components accounted for 75.2% (PC1 = 57.9% and PC2 = 17.3%) and 59.8% (PC1 = 35.8% and PC2 = 24.0%) of the cumulative percent variability in soil health at 0–5 cm and 5–15 cm depth, respectively (Fig. 2a; Supplementary Fig. 4a). The soil health score (Cornell Soil Health Assessment Scoring) was affected

by management, warming, and their interaction (*P* < 0.05; Supplementary Table 2). The response of soil health to warming varied depending on soil management (*P* < 0.05; Supplementary Table 2). Warming increased the soil health score by 6.3% and 8.1% at 0–5 cm and 5–15 cm soil depth under conservation agriculture, but only increased soil health score by 5.2% at 5–15 cm depth under conventional agriculture (*P* < 0.01; Fig. 2b; Supplementary Fig. 4b; Supplementary Table 2). The soil health score under conservation agriculture was 21.5% and 7.1% greater than conventional agriculture at 0–5 cm and 5–15 cm depth in ambient conditions, respectively (Fig. 2b; Supplementary Fig. 4). Warming amplified the advantages of conservation agriculture in terms of soil health, and resulted in a 31.4% and 10.1% greater soil health score than conventional agriculture at 0–5 cm and 5–15 cm depth, respectively (Fig. 2b; Supplementary Fig. 4b).

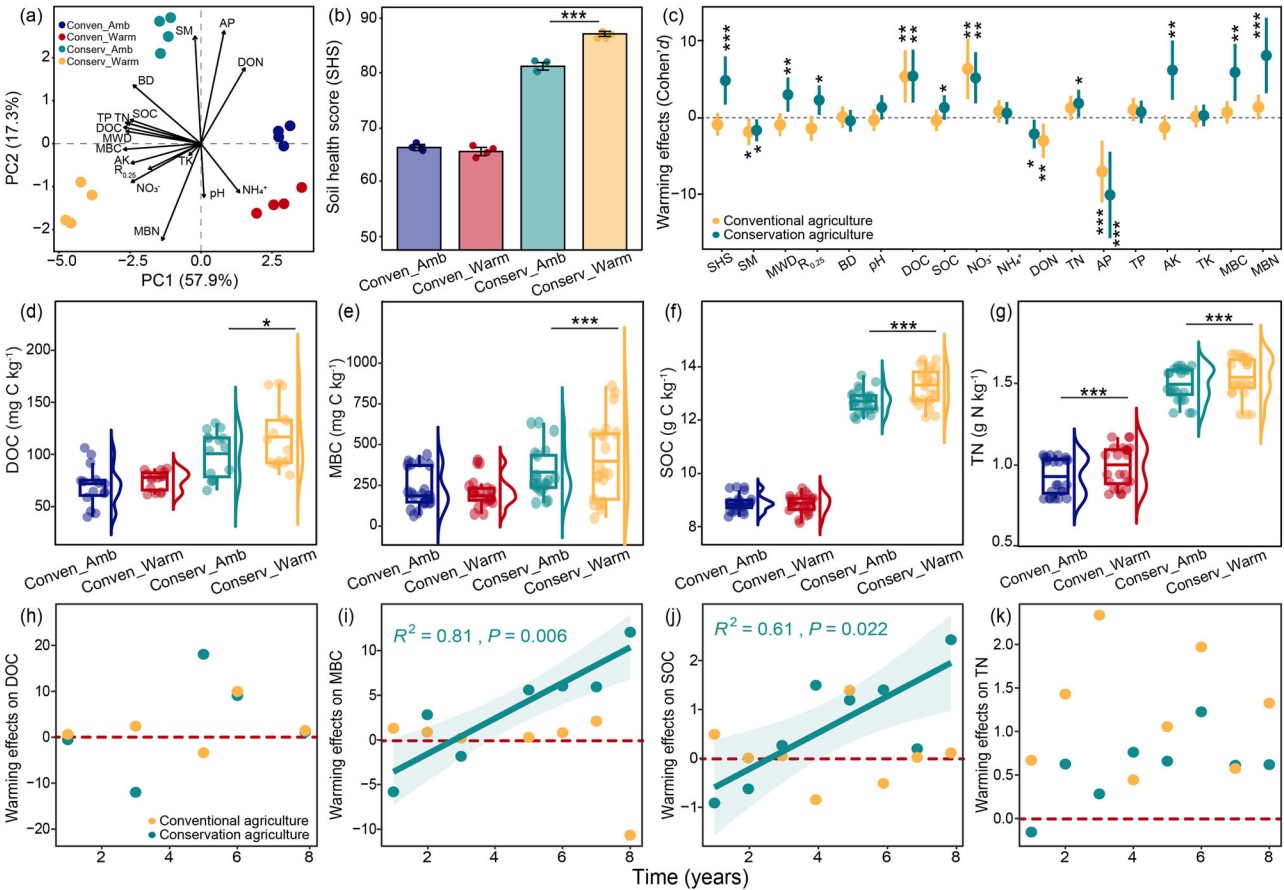

**Fig. 2 | Effects of warming and management on soil attributes and soil health score at 0-5 cm soil depth. a** Two dimensions of Principal Component Analysis (PCA) for eigenvalues of the seventeen soil attributes, including MWD, $R_{0.25}$, SM, BD, pH, DOC, SOC, $NH_4^+$-N, $NO_3^-$-N, DON, TN, AP, TP, AK, TK, MBC, and MBN. All parameters were analyzed based on soil sampling in 2020 ($n = 4$ independent soil samples per treatment). **b** Effects of warming and management on soil health score. Data are presented with mean values ± s.e.m. Statistical analysis was performed using two-way ANOVA analysis. Soil health score was evaluated based on soil sampling in 2020 ($n = 4$ independent soil samples per treatment). **c** Estimates of the warming effects on soil attributes depending on management. The effect size was estimated by Cohen' $d$. Data are presented with mean values ± s.e.m. Statistical analysis was performed using two-way ANOVA analysis ($n = 4$ independent soil samples per treatment). **d–g** Average of key soil attributes over eight years determined using a linear mixed model. Statistical significance is based on Wald type II $\chi^2$ tests ($n = 24$ independent soil samples per treatment). Boxplots display the mean (horizontal line), the 25th and 75th percentiles (colored box), the minimum and maximum (whiskers). **h–k** Temporal changes of warming effects on key soil properties over time depending on management. The effect size was estimated by Cohen' $d$. The solid line represents the significant linear regression ($P < 0.05$), and the gray shading indicates the 95% confidence interval (i.e., error bands represent slopes ± 95% confidence intervals). All reported $P$ values result from two-sided statistical tests where *$P < 0.05$, **$P < 0.01$, and ***$P < 0.001$. Conserv-Amb, conservation agriculture without warming; Conserv-Warm, conservation agriculture with warming; Conven-Amb, conventional agriculture without warming; Conven-Warm, conventional agriculture with warming. MWD, mean weight diameter; $R_{0.25}$, aggregate content with particle size larger than 0.25 mm; SM soil moisture, BD bulk density, DOC dissolved organic carbon, SOC soil organic carbon, $NH_4^+$-N ammonium, $NO_3^-$-N nitrate, DON dissolved organic nitrogen, TN total nitrogen, AP available phosphorus, TP total phosphorus, AK available potassium, TK total potassium, MBC microbial biomass carbon, MBN microbial biomass nitrogen. Source data are provided as a Source Data file.

We also calculated the warming effects on individual soil indicators that contributed to the soil health score between paired warmed and ambient plots under the two management systems (Fig. 2c–g; Supplementary Fig. 3; Supplementary Fig. 4c–g; Supplementary Table 2). Warming enhanced mean weight diameter (MWD) and the proportion of soil aggregates larger than 0.25 mm ($R_{0.25}$) under conservation agriculture but not conventional agriculture, indicating better water infiltration, storage and supply (Fig. 2c; Supplementary Figs. 3–5). Warming also promoted nutrient cycling, storage and supply under conservation agriculture, characterized by increased DOC, SOC, TN, AK, and $NO_3^-$-N concentration under warming (Fig. 2c; Supplementary Figs. 3–5). Conservation agriculture increased MBC and MBN and suggested greater microbial activity under warming (Fig. 2c; Supplementary Figs. 3–5). We calculated the weighting values of individual soil properties and recognized DOC, MBC, SOC, and TN as the key contributors to the soil health score (Supplementary Fig. 3).

Increased DOC, MBC, SOC, and TN were stimulated by warming and collectively improved the soil health of conservation agriculture (Fig. 2c–g; Supplementary Fig. 3; Supplementary Fig. 4c–g). We further assessed the warming effect on DOC, MBC, SOC, and TN over eight years (Fig. 2d–k; Supplementary Fig. 4d–k). Across the eight-year study, the average soil DOC was increased by warming under both conservation and conventional agriculture ($P < 0.05$; Fig. 2d-g; Supplementary Fig. 4d-g; Supplementary Table 1), but changes in MBC and SOC in response to warming varied depending on soil management at 0–5 cm depth ($P < 0.05$; Fig. 2d, f; Supplementary Fig. 4d, f; Supplementary Table 1). Warming increased MBC and SOC under conservation agriculture but not under conventional agriculture ($P < 0.05$; Fig. 2e, f; Supplementary Table 1). Moreover, the positive effects of warming on MBC and SOC strengthened with time under conservation agriculture ($P < 0.05$; Fig. 2h–k), indicating a cumulative benefit of conservation agriculture over time for these soil carbon pools.

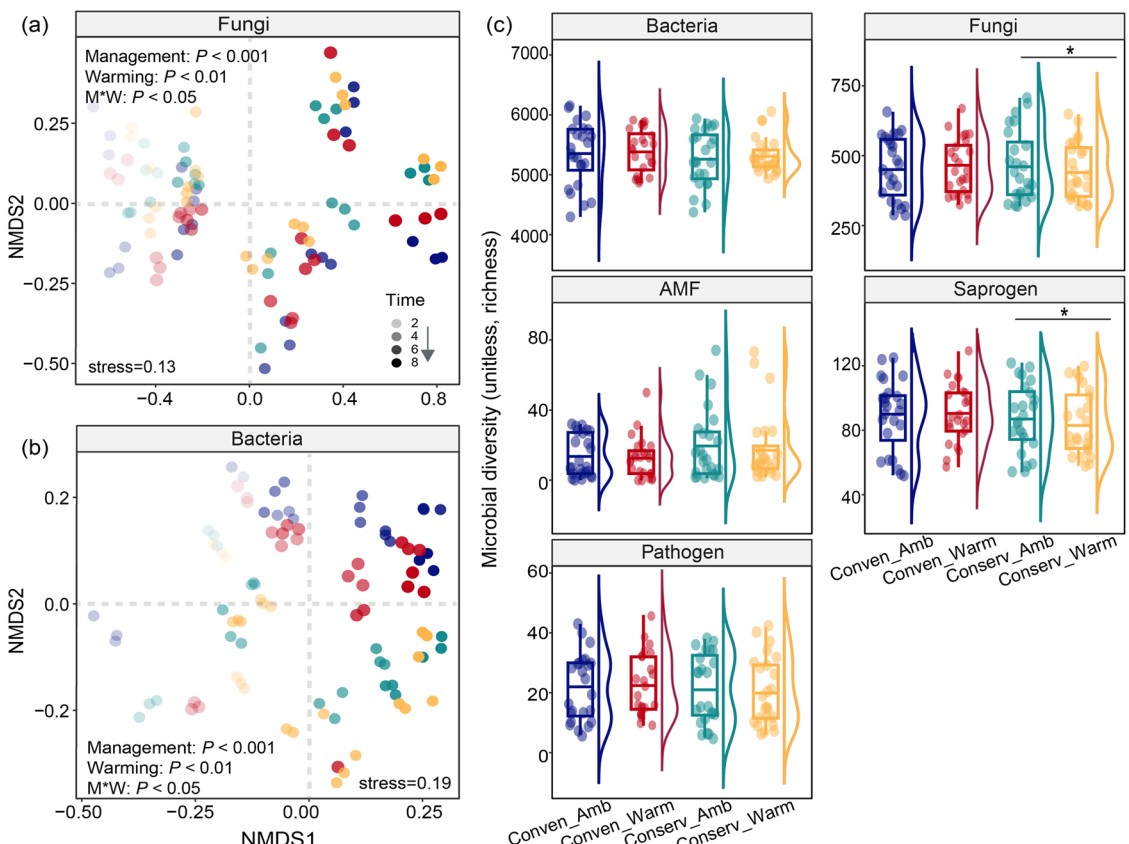

**Fig. 3 | Effects of warming and management on microbial diversity and community composition at 0-5 cm soil depth. a, b** Nonmetric multidimensional scaling (NMDS) ordination of soil fungal and bacterial communities based on the Bray-Curtis dissimilarity. Statistical analysis was performed using nested permutational multivariate analysis of variance (nested PERMANOVA) analysis. **c** Effects of warming and management on microbial richness of soil total fungi, fungal guild, and bacteria. Boxplots display the mean (horizontal line), the 25th and 75th percentiles (colored box), the minimum and maximum (whiskers). Data were analyzed based on eight sampling years ($n = 24$ independent soil samples per treatment). Statistical analysis was performed using linear mixed model with sampling time as random factors. Statistical significance is based on Wald type II $\chi^2$ tests. All reported $P$ values result from two-sided statistical tests where *$P < 0.05$, **$P < 0.01$, and ***$P < 0.001$. Conserv-Amb, conservation agriculture without warming; Conserv-Warm, conservation agriculture with warming; Conven-Amb, conventional agriculture without warming; Conven-Warm, conventional agriculture with warming. Source data are provided as a Source Data file.

## Effects of warming and management on microbial diversity and community composition

Soil fungal and bacterial diversity and community composition were determined across eight years (Fig. 3; Supplementary Fig. 6; Supplementary Tables 2–3). Our results based on NMDS ordinations and nested PERMANOVA analysis indicated that soil fungal and bacterial communities at 0-5 cm depth were affected by management ($R^2 = 0.089–0.103$, $P < 0.001$), warming ($R^2 = 0.014–0.017$, $P < 0.01$), and their interaction ($R^2 = 0.009–0.010$, $P < 0.05$) (Fig. 3a; Supplementary Table 3). The microbial diversity over eight years, as estimated by richness, was further assessed through linear mixed modelling (Supplementary Table 1). The soil bacterial richness was not affected by management, warming, and their interaction (Supplementary Table 1). The total soil fungal richness was influenced by interactions between warming and management ($P < 0.05$; Fig. 3c; Supplementary Table 1), indicating differential responses of the soil fungal community to warming under conservation and conventional agriculture. Warming decreased fungal richness by 4.1% compared to the no-warming treatment under conservation agriculture ($P < 0.05$), but not under conventional agriculture (Fig. 3c). Specifically, the richness of fungal functional guilds was influenced by management, but not warming (Fig. 3c; Supplementary Table 1). Conservation agriculture supported the greater richness of AMF ($P < 0.001$; Supplementary Table 1) and lesser richness of saprogens and pathogens ($P < 0.05$; Supplementary Table 1) at 0-5 cm soil depth (Fig. 3c).

Warming decreased the richness of saprogens under conservation agriculture only ($P < 0.05$; Supplementary Table 1) (Fig. 3c).

Soil fungal and bacterial communities in the 5–15 cm soil depth were affected by management ($R^2 = 0.061–0.072$, $P < 0.001$), warming ($R^2 = 0.024$, $P < 0.01$), and their interaction ($R^2 = 0.023$, $P < 0.05$) (Supplementary Fig. 6b; Supplementary Table 3). Bacterial richness was only affected by management and was greater under conventional agriculture ($P < 0.01$; Supplementary Fig. 6c; Supplementary Table 1). Total fungal richness was not affected by management, but richness of AMF and saprogen varied between management systems (Supplementary Fig. 6b; Supplementary Table 1). Conservation agriculture supported more AMF ($P < 0.001$; Supplementary Table 1) and less saprogens ($P < 0.05$; Supplementary Fig. 6c; Supplementary Table 1). Warming decreased total fungal richness ($P < 0.01$) and decreased saprogens ($P < 0.001$) and pathogens ($P < 0.01$) under conventional agriculture (Supplementary Fig. 6c; Supplementary Table 1).

## Soil health and the microbiome contributed to wheat yields under conservation agriculture with warming

We selected the best soil properties to predict crop yield using linear mixed models (Fig. 4; Supplementary Figs. 7–8). The MBC and richness of saprogens at 0–5 cm soil depth, and fungal community composition at 5–15 cm soil depth, were most closely linked to wheat yields under conservation agriculture (Fig. 4a, b; Supplementary Fig. 7). There were corresponding strong positive correlations

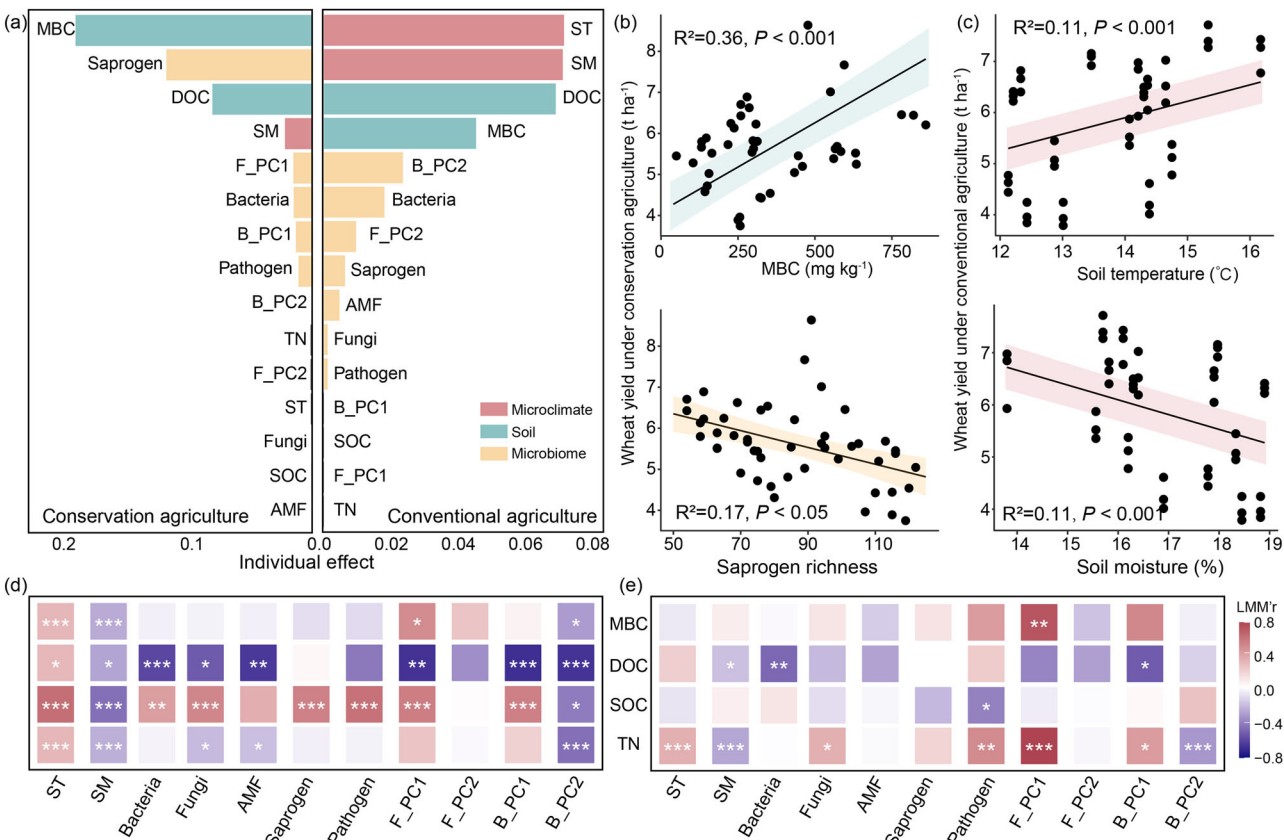

**Fig. 4 | Effects of abiotic and biotic factors on winter wheat yield and key indicators of soil health of 0-5 cm soil depth. a** Individual effect of the model predictors to winter wheat yield under conservation and conventional agriculture, respectively. Red, green and orange colors represent microclimate, soil, and microbial factors, respectively. **b, c** Correlations between wheat yield and key predictors under conservation and conventional agriculture. Solid line in each panel shows significant model fit using linear mixed-effect regression ($P < 0.05$), and the shading around the fitted line represents the 95% confidence intervals (i.e., error bands represent slopes ± 95% confidence intervals). **d, e** Correlations between soil microclimate and microbial diversity and key indicators of soil health at 0–5 cm soil depth under conservation and conventional agriculture. The color denotes the correlation coefficient determined by the linear mixed-effects model. Statistical significance is based on Wald type II $\chi^2$ tests. ST soil temperature, SM soil moisture, DOC dissolved organic carbon, MBC microbial biomass carbon, SOC soil organic carbon, TN total nitrogen. Bacteria, Fungi, AMF, Saprogen, and Pathogen indicated richness of microbial group. F_PC1, F_PC2, B_PC1, and B_PC2 indicated fungal and bacterial community composition, respectively. All reported $P$ values result from two-sided statistical tests where *$P < 0.05$, **$P < 0.01$, and ***$P < 0.001$. Source data are provided as a Source Data file.

between wheat yields and MBC, but negative correlations with saprogen richness (Fig. 4b). Soil MBC stimulated by warming, reductions in saprogen richness and variations of the fungal community contributed most to wheat yields under conservation agriculture (Fig. 4b; Supplementary Fig. 7). In addition, the richness of soil bacteria and pathogens also affected wheat yields under conservation agriculture (Fig. 4a; Supplementary Fig. 7). In contrast, the soil microclimate (temperature and moisture) and DOC were most relevant to wheat yield under conventional agriculture (Fig. 4a, c; Supplementary Fig. 7). Summer maize yields were also affected by soil properties, but the measured soil variables did not lead to changes in maize yields under warming (Supplementary Fig. 8). In conclusion, crop yields under conservation agriculture were more influenced by the soil microbiome (including MBC, fungal richness, and fungal community composition) than under conventional agriculture.

The key indicators of soil health were also affected by soil microbial richness and community composition (Fig. 4d, e; Supplementary Fig. 9), especially under conservation agriculture. Soil MBC and SOC were more relevant to soil fungal richness or community composition compared with bacteria under conservation agriculture (Fig. 4d, e; Supplementary Fig. 9). The intense relationship between soil fungi and carbon emphasized the microbially-mediated

mechanisms that underlie soil carbon accrual observed under conservation agriculture.

## Discussion

This study provides rare empirical evidence for the advantages of conservation agriculture to improve soil health and sustain crop yields in a warming world and emphasizes the potential of conservation agriculture in long-term food security. The benefits of conservation agriculture for multiple soil functions have been reported[4,7], but the interactive effects of management practices and climate warming on the maintenance of soil health and crop productivity have not been tested before in arable land in situ. Both crop yields and key indicators of soil health (e.g., SOC and MBC) strengthened over time in response to warming, but only under conservation agriculture. These beneficial responses were tightly linked to soil fungal richness and community composition. In summary, we provide evidence that conservation agriculture can support greater soil health without compromising crop yields under long-term warming compared to conventional agriculture and reveal a key role of the soil microbiome.

The improvement of soil health to deliver resilient 'climate-smart' farming systems is recognized as a fundamental strategy that is vital to mitigate and adapt to the adverse effects of climate warming to maintain global food security[1,52]. We found that conservation

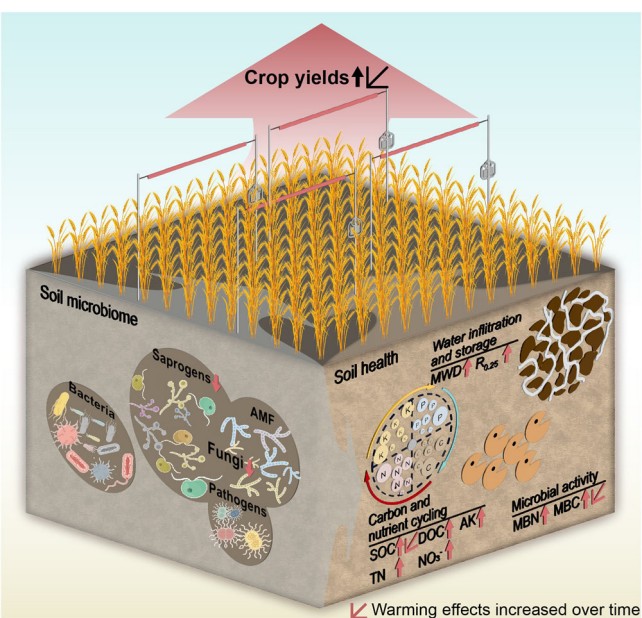

**Fig. 5 | Conceptual model illustrating shifts of soil health, microbial diversity, and crop yield under conservation agriculture response to warming.** Experimental warming triggered shifts in soil fungal richness and community composition and resulted in an improved soil health under conservation agriculture, including increased water infiltration and storage, carbon and nutrient cycling, and microbial activity. The improvement in soil health and shifts in soil fungal diversity contributed to higher crop yields under conservation agriculture. The up or down arrows showed increased or decreased soil health indicators response to warming under conservation agriculture. The linear trends indicated warming effects on crop yields and soil health indicators over time under conservation agriculture. MWD, mean weight diameter; $R_{0.25}$, aggregate content with particle size larger than 0.25 mm; DOC dissolved organic carbon, SOC soil organic carbon, $NO_3^-$ nitrate, AK available potassium, MBC microbial biomass carbon, MBN microbial biomass nitrogen. The soil microbiome drawing elements were produced using figdraw.com.

agriculture improved a suite of soil properties involved in key soil functions (water infiltration and storage, carbon and nutrient cycling, and microbial activity) after eight years of warming (Figs. 2, 5). Soil C is considered a principal indicator of soil health and agronomic sustainability due to its impact on physical, chemical and biological properties[29,53,54]. Warming can promote plant carbon inputs (includes root biomass, exudation, and crop resides) to soil, thereby increasing substrates for the soil microbial community[17,55]. In our study, conservation agriculture increased all soil C pools (DOC, MBC, and SOC) compared to conventional agriculture (Fig. 2). Moreover, we observed positive warming effects on SOC and MBC under conservation agriculture that increased linearly with experimental duration (Fig. 2i, j), indicating a long-term advantage of conservation agriculture in supporting microbial activity and contributing to soil carbon accrual[17,18]. Reduced soil disturbance under conservation agriculture promotes soil aggregate formation and stability indicated by MWD and $R_{0.25}$ (Fig. 2c). Soil aggregate stability is considered a useful indicator of soil structure that supports multiple soil functions including water transport and storage and the provision of stable microbial habitats conducive to the preferential accumulation of microbial biomass and necromass formation and further SOC accrual[56,57]. The increased soil aggregate stability can enhance water conservation and microbial variables, that directly facilitate nutrient cycling and soil carbon storage[56,57]. The tighter correlations between soil fungal richness and community composition, rather than bacteria (Fig. 4), suggested soil fungi may be the major contributor to SOC formation and

accumulation under conservation agriculture. Collectively, conservation agriculture is a sustainable management and performs better in adapting climate warming than conventional agriculture in Northern China Plain. Previous studies found that conservation agriculture accumulates soil C at the surface rather than the whole soil profile[58,59], or even changed distribution of C in the soil profile instead of increasing the total SOC[14,60]. Yet our long-term monitoring only focuses on C accumulation at surface soil (0–15 cm) and may overestimate benefits in C sequestration over the whole soil profile. The changes of SOC in the deeper soil profiles need to be pay more attention in the future research.

Globally, crop yields under conservation agriculture are estimated to be 2.5% less than those of conventional agriculture[13], but we did not observe reductions in wheat or maize under conservation agriculture in either ambient or warming conditions (Fig. 1; Supplementary Fig. 1). Previous studies in the tropics have reported warming decreased crop yields[7,25]. We found warming increased wheat yields by 9.3% and 11.2% under conservation and conventional agriculture (Fig. 1a), respectively, on temperate semi-arid climate of the North China Plain. Here, warming extends the growing season allowing earlier planting and later harvests, and can reduce cold injury and frost damage in seedlings[51,61], which directly improves crop growth and productivity[61]. In accordance with predictions that crop yields would be enhanced by warming in middle-to-high latitude regions[7,25], we observed a positive relationship between wheat yields and soil temperature under conventional agriculture (Fig. 4a). By comparison, increased wheat yields under conservation agriculture and warming were most strongly associated with improved soil microbial quality indicated by microbial biomass carbon (MBC). Positive correlations between crop productivity and MBC are reported[62,63] because the microbial biomass is not only a labile carbon pool but also a nutrient reservoir[64,65]. The soil microenvironment (soil and temperature) was also relatively important for wheat yields under conservation agriculture with warming (Fig. 4a, b) because no tillage and residue retention moderate soil temperature and retain soil moisture (Fig. 1). Importantly, we also observed that the warming-induced positive effects on wheat yield strengthened with time (Fig. 1b). Warming had no effect on maize yields regardless of management type (Supplementary Fig. 1); the maize growing season is in summer (June–September) and C4 physiology is adapted to warmer temperatures and drier conditions[13]. The results for wheat emphasize the vital role of soil health in reducing the sensitivity of production of this key cereal crop to long-term changes in climate and reducing climate-driven yield variability[1,66]. In consideration of undifferentiated wheat yield and higher maize yield compared to conventional agriculture both under ambient and warming conditions, we assumed that conservation agriculture performed better in productivity in a warmer world.

Soil microbiota are prominent actors in SOC accrual, decomposition and nutrient cycling[67–70], and consequently strongly influence agroecosystem services[15,71], so it imperative to assess their response to long-term warming in situ. Reduced soil biodiversity is reported in some natural ecosystems under warming, mainly due to environmental filtration by temperature and soil moisture reduction[47,49,72,73]. We observed a reduction of total fungal richness, but not of bacteria, in soil from the 0–5 cm depth under conservation agriculture (Fig. 3; Supplementary Fig. 6), corresponding to previous studies showing greater sensitivity of fungi to warming compared to bacteria[74,75]. The decrease of fungal richness was predominantly a consequence of a decrease in saprogen richness under conservation agriculture with warming (Fig. 3; Supplementary Fig. 6). Less saprogen richness may be linked to slower rates of organic matter decomposition[76] that contributes to SOC accrual in conservation agriculture under warming (Fig. 2), that supports yield increase through the net effect on soil health (Fig. 4a, b). Soil AMF can enhance plant nutrient uptake and reduce plant stress[71], and were more diverse under conservation

agriculture with warming (Supplementary Table 1). Soil-borne plant pathogens were less diverse under conservation agriculture with warming, providing further advantages for crop production and disease management[13,28,41]. Overall, the direct and indirect effects of conservation agriculture on the fungal community and plant pathogens were related to enhanced crops yield and soil health.

In summary, this study provides empirical evidence for the potential benefits of conservation agriculture for long-term sustainable food production because improved soil health improves resilience to the effects of climate warming through its effects on physical, chemical and biological soil properties. However, we recognize that local climatic conditions are an important driver of soil health and crop yield[7] and careful regional assessments are needed when considering the potential consequences of adopting conservation agriculture[7,13]. We propose that our findings may be generalized to other regions where water does not limit productivity (e.g., irrigated regions). As illustrated in Fig. 5, the combination of conservation agriculture and experimental warming for eight years on the temperate semi-arid climate of the North China Plain stimulated an increase in soil health indicators indicating improved water infiltration and storage (soil aggregate stability), carbon and nutrient cycling, and microbial activity. The improvement in soil health and shifts in soil fungal diversity (less saprogen richness, increased AMF and less plant pathogens) supported similar crop yields to conventional agriculture. These findings emphasize the potential cumulative benefits of conservation agriculture with time and strengthen the evidence for conservation agriculture as 'climate-smart' management tool to adapt to climate warming and ensure good security by improving soil health.

## Methods

### Study sites
This study is based on a long-term climate change field trial located at North China Plain at Yucheng Comprehensive Experiment Station of Chinese Academy of Science (36° 50' N, 116° 34' E, elevation is 20 m). The study region has a temperate semi-arid climate with an annual mean temperature of 13.6 °C, and annual mean precipitation of 575 mm with 70% occurring between June and September. The soil type of this site is Calcaric Fluvisol (FAO-UNESCO system) with typical soil texture 12% sand, 66% silt, 22% clay, and a mean pH of 7.1. The experiment was conducted at a crop rotation system (winter wheat (*Triticum aestivum* L.)-summer maize (*Zea mays* L.)) since 2010.

### Experimental design and soil sampling
The system included four treatments: conventional agriculture with and without warming (Conven-Warm and Conven-Amb), conservation agriculture with and without warming (Conserv-Warm and Conserv-Amb). A randomized complete block design randomized with four replicates was employed. The warmed soils (+2 °C) were continuously heated (since February, 2010) using an MSR-2420 infrared heater (Kalglo Electronics Inc, Bethlehem, PA, USA) suspended approximately 3 m above the ground to achieve a surface soil warming of 2 °C, which is predicted by IPCC greenhouse gas scenarios rates for northern China[51]. The control plots (i.e., without warming) were treated with a 'dummy' infrared heater to simulate the shading effects of the heaters. There was a 5 m border between adjacent blocks and at least 10 m between the plots to avoid heating the control plots by the infrared radiators.

The size of each plot was 2 m × 2 m. Winter wheat was seeded between 10 and 15 October and harvested during the first 10 days of June. Then summer maize was seeded 5 days later and harvested during the first 10 days of October. After harvest, in the conventional agriculture treatment, the residues were removed and the soil was cultivated with a rotary tiller annually. In the conservation agriculture treatment, all residues were chopped to approximately 5 cm in length and retained on the soil surface, and adopted no tillage. The

conventional and conservation agriculture treatments had the same total $N$ application rate, but part of the $N$ application was from different sources. The total $N$ application for conventional agriculture and conservation agriculture treatments in winter wheat growing seasons was 285 kg N ha$^{-1}$ yr$^{-1}$ following optimal application of fertilization[77,78]. For conservation agriculture, the $N$ input was 112.5 kg ha$^{-1}$ of mineral fertilizer, 124.5 kg ha$^{-1}$ of urea, and 48 kg ha$^{-1}$ residue N. For conventional agriculture, the N input was 112.5 kg ha$^{-1}$ of mineral fertilizer, and 172.5 kg ha$^{-1}$ of urea. The total N application for conventional agriculture and conservation agriculture in summer maize growing seasons was 207 kg N ha$^{-1}$ yr$^{-1}$. For conservation agriculture, the N input was 175 kg ha$^{-1}$ of urea and 32 kg ha$^{-1}$ residue N. For conventional agriculture, the $N$ input was 207 kg ha$^{-1}$ of urea. All other management procedures were the same for conventional agriculture and conservation agriculture.

Three composite soil samples from 0–5 and 5–15 cm soil depths were collected for each plot after harvest of winter wheat from 2010 to 2019. Composite samples were collected by hand augur consisting of five randomly selected soil cores in each plot, then mixed together to make a composite sample. All samples were passed through a 2 mm sieve for subsequent analysis. Soil temperature (ST) and volumetric soil moisture (SM) at 0–15 cm depth was monitored by PT100 thermocouples and FDS100 soil moisture sensors (Unism Technologies Incorporated, Beijing), respectively.

### Assessing crop productivity
Crop yields were determined through harvest of each plot by hand. At physiological maturity of each crop, 1 m$^2$ of crop was harvested for each plot and the sample was threshed using a static machine. Crop yield was measured after air-drying to a constant value. Plot yield was recalculated on a dry matter basis and the harvested area was used to determine yield per unit area (t ha$^{-1}$).

### Assessing soil health
Seventeen soil health indicators were measured as variables to proxy soil physical, chemical, and biological attributes in 2020 to evaluate cumulative changes after long-term experimental warming. Soil samples (0–5 cm and 5–15 cm depth) were collected after the harvest of winter wheat to measure soil properties including: aggregate mean weight diameter (MWD), aggregate content with particle size larger than 0.25 mm (R$_{0.25}$), soil moisture (SM), bulk density (BD), pH, dissolved organic carbon (DOC), soil organic carbon (SOC), NO$_3^-$-N, NH$_4^+$-N, dissolved organic nitrogen (DON), total nitrogen (TN), available phosphorus (AP), total phosphorus (TP), available potassium (AK), total potassium (TK), microbial biomass carbon (MBC) and microbial biomass nitrogen (MBN). Soil samples (0–5 cm and 5–15 cm depth) were collected after harvest of winter wheat to measure soil properties.

Soil aggregate mean weight diameter (MWD) and content with particle size larger than 0.25 mm (R$_{0.25}$) was obtained using wet sieving. Soil pH was measured using a 1:2.5 (w/v) soil to 0.01 M CaCl$_2$ ratio with a glass electrode. Soil bulk density (BD) was obtained from oven dry mass relative to the sample volume. The subsamples for SOC and TN analysis were air-dried at room temperature and determined using elemental analysis (Vario EL III, Elementar, Germany) and Kjeldahl digestion[79]. The concentration of DOC and dissolved nitrogen (DN) was measured according to Jones and Willett[80]: field-moist soil samples (equivalent to 15 g oven-dried soil) were extracted with 60 ml of 0.05 mol/L K$_2$SO$_4$ (soil to solution ratio 1:4) for 1 h, and the extract was passed through a 0.45 mm membrane filter and analyzed for DOC and DN using a Multi 3100 N/C TOC analyzer (Analytik, Germany). Soil NH$_4^+$ and NO$_3^-$ were determined by extraction with 2 mol L$^{-1}$ KCl. The DON concentration was calculated as DN minus NH$_4^+$-N and NO$_3^-$-N content. Soil TP was measured by colorimetric analysis after digestion with sulfuric acid and perchloric acid. Soil AP was determined by the Olsen method. Soil TK and AK were determined by flame photometer. The

content of MBC and MBN was determined according to method of Wu et al.[81]: fresh soil samples were fumigated with chloroform (48 h) and subsequent extracted with 0.5 M $K_2SO_4$ (soil to solution ratio 1:4) and organic carbon and nitrogen in the extractant were measured using an elemental analyzer (Vario PYRO Cube, Hanau, Germany).

We used the Cornell Soil Health Assessment (CSHA) scoring method to calculate soil health scores[82]. The total data set, including 17 soil indicators (physical attributes: MWD, $R_{0.25}$, SM, BD; chemical attributes: soil pH, SOC, DOC, TN, DON, $NH_4^+$-N, $NO_3^-$-N, AP, TP, AK, TK; biological attributes: MBC and MBN), were normalized as individual CSHA scores[53,82]. The weighting value of individual soil indicators was based on PCA of all soil indicators (Supplementary Fig. 2), representing the sum of the eigenvectors derived from the first three principal components, which were selected based on Kaiser's cut-off (eigenvalues > 1). These first three principal components cumulatively accounted for over 70% of variance, capturing most of the variation among the soil indicators. The overall soil health score (%) was computed as a weighted average of all individual CSHA scores as follows[53,82]:

$$Soil\ health\ score = \frac{A_1 \times w_1 + A_2 \times w_2 + \cdots + A_n \times w_n}{w_1 + w_2 + \cdots + w_n} \tag{1}$$

where $A$ is the CSHA score for each individual soil indicator, and $w$ is the weighting value of the soil indicators (Supplementary Fig. 2).

### Amplicon sequencing and soil biodiversity

To discern whether climate warming and management practices affect soil bacterial and fungal diversity, soil DNA was extracted from 0.25 g soil with PowerSoil Isolation kit (MoBio Laboratories, Carlsbad, CA, USA) following the modified manufacturer's recommendations. DNA concentrations were quality checked (NanoDrop One; Thermo Fisher Scientific, Waltham, MA) and quantified (PicoGreen assay; Quant-iT™ PicoGreen® dsDNA Reagent, Life Technologies). Finally, all DNA samples were stored at −80 °C until sequencing analysis.

The library construction and sequencing of the 16S rRNA gene and ITS gene were performed. The universal primer sets, 515 F (5′-GTGCCAGCMGCCGCGC-3′) and 907 R (5′-CCGTCAATTCMTT-TRAGTTT-3′) targeting the V4-V5 hypervariable regions of the bacterial 16S rRNA gene, and gITS7F (5′-GTGARTCATCGARTCTTTG-3′) and ITS4R (5′-TCCTCCGCTTATTGATATGC-3′) for the fungal IT, were used in this study. After PCR and purification, a DNA library was constructed and run on an Illumina Hiseq platform at MagiGene Biotechnology Co., Ltd. (Guangzhou, China).

The raw sequence data were processed with the USEARCH (version 11.0.667)[83] software for quality filtering and assembling of pair-end reads. Strict quality control steps were applied to the sequencing data. First, assembled contigs without exact match to one of the barcodes sets or primers (degenerate bases were not taken into consideration) were discarded. Subsequently, the remaining sequences were clustered into high accuracy exact amplicon sequence variants (ASVs) with the UNOISE[84]. Singletons that found only once across all samples were removed from subsequent analyses. Taxonomic classification of each ASV of bacteria and fungi was determined using the Ribosomal Database Project (RDP) classifier with a confidence threshold of 0.5 against the SILVA[85] and the UNITE[86] database, respectively. ASVs that were not classified into bacteria or fungi were removed. To normalize samples to the same total read abundance, an average of 26,000 and 20,000 sequence reads per sample were randomly selected (resampled) for each sample obtained for 16S rRNA gene and ITS, respectively. Bacterial and fungal richness was calculated based on the phylogenetic trees and ASV tables using the R package *picante*.

We obtained the relative abundance of potential arbuscular mycorrhizal mutualists, saprotrophs, and plant pathotrophs from amplicon sequencing by parsing the soil fungal phylotypes using FUNGuild database[87]. Fungi representing <1% of the ASVs were not considered. Only highly probable and probable guilds were used in these analyses.

### Statistical analyses

Linear mixed models (LMMs) were used to assess the effects of warming and management practices on soil properties, biodiversity, and crop yield measured repeatedly across eight years[47]. In the LMMs, warming (0 for ambient temperature and 1 for warming) and management practices (0 for conservation agriculture and 1 for conventional agriculture) were considered as fixed effects, while sampling time (year) was termed as random intercept effects (y ~ warming × management + (1 | year)). Wald Type II $\chi^2$ tests were used to calculate the $P$ values from the LMMs. The *lme4* R package was used to implement LMMs. Differences between four treatments (Conven-Amb, Coven-Warm, Conserv-Amb, and Conserv-Warm) were estimated through pairwise comparison with estimated marginal means. The *emmeans* R package was use to implement pairwise comparison. Two-way ANOVA was used to evaluate the effects of warming and management practices on soil parameters and soil health score. To evaluate the effects size of warming on crop yield, soil health and each soil indicator, we calculated Cohen's *d* from different management practices by comparing them against the common control without warming through *effsize* R package. The temporal changes of warming effects on each soil indicator and crop yield were estimated through a linear regression.

Non-metric multidimensional scaling (NMDS) based on Bray-Curtis dissimilarity and nested permutational multivariate analysis of variance (nested PERMANOVA) was used to illustrate the effects of warming and management practices on soil microbial community composition through *vegan* R package. To link the soil microbial diversity and community composition to soil health and crop yield, the correlations were tested using the linear mixed-effects model, in which sampling year were termed as random intercept effects. The correlation r was represented by the regression coefficients in the LMMs, and Wald type II $\chi^2$ tests were used to calculate the $P$ values from the LMMs. Hierarchical partitioning analysis to calculate the individual contributions of each predictor was conducted with the *glmm.hp* R package to determine the relative importance of each environmental factor[88]. All the analyses were performed in the R 4.3.0.

### Reporting summary

Further information on research design is available in the Nature Portfolio Reporting Summary linked to this article.

## Data availability

The DNA sequences of the 16S rRNA gene and ITS amplicons in this study have been deposited in the National Center for Biotechnology Information (NCBI) under project accession numbers PRJNA996529. Silva database is available at https://www.arb-silva.de/. UNITE database is available at https://unite.ut.ee/. Source data are provided with this paper.

## Code availability

The analysis code that supports the findings of this study is available on GitHub (https://github.com/bio-carbon/warming_soil_health)[89].

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

## Acknowledgements

This study was supported by the National Natural Science Foundation of China (grant nos. U23A20158, Z.L.C., and 32071629, J.T.), National Key R&D Program of China (2023YFD1901500, J.T.), 2115 Talent Development Program of China Agricultural University and Beijing Advanced Disciplines, F.S.Z., Z.L.C., J.B.Z., J.T., and the China Postdoctoral Science Foundation (2022M71339, J.L.T.). We thank the drawing tools provided by Figdraw.

## Author contributions

All authors contributed intellectual input and assistance to this study and manuscript preparation. T.J., Z.L.C., and M.D.B. designed the original concept and experiment strategy. J.L.T. and R.X.H. collected the data. J.L.T. and J.T. analyzed the data. R.X.H. carried the field experiment. J.L.T., J.T., J.A.J.D., G.Y.Z., Y.K., J.B.Z., Z.L.C., F.S.Z., and M.D.B. drafted the manuscript and all authors involved in critical revisions and gave final approval for publication.

## Competing interests

The authors declare no competing interests.

## Additional information

**Supplementary information** The online version contains Supplementary Material available at https://doi.org/10.1038/s41467-024-53169-6.

