## [Transparent Peer Review file · Nature Communications]

Conservation agriculture improves soil health and sustains crop yields after long-term warming

Corresponding Author: Dr Manuel Delgado-Baquerizo

Version 0:

Reviewer comments:

Reviewer #1

(Remarks to the Author)

General comments:

The manuscript entitled "Conservation agriculture supports ecosystem multifunctionality after decade-long warming" presents an interesting study, which tackle two timely and relevant issues of agricultural production that are the holistic evaluation of different cropping systems on various ecosystem functions and their adaptation potential in face of climate change. In their study, the authors compare the impacts of conservation tillage (agriculture) and traditional cropping with tillage on several ecosystem properties over 10 years in a field experiment and how these impacts are affected by climate warming. Overall, the study and the field experiment are well designed and I acknowledge the huge efforts invested in conducting the warming experiment and the data collection. I also appreciate the analysis of synergies and trade-offs across the studied variables (even if I doubt about their real implications).

However, they are too many drawbacks and limitations in term of concept (EMF and cropping system classification), methodology (assessed variables as proxy for functions) as well as presentation and interpretation of the results in order to support this manuscript for publication (see below).

I suggest the authors to rethink and narrow the framework of the study. Given the data availability, I would focus more on the effects of the experimental treatments on soil (microbiological) quality and how particular soil processes are impacted by warming. I also strongly suggest to not only present and discuss the data on a relative scale but also reflect actual absolute changes and their real implications for (soil) ecosystem functioning.

Concept:

On ecosystem multifunctionality:

The importance of assessing the performance of agricultural production on various aspect of the ecosystem, other than productivity, is now broadly recognize and acknowledged. An important and increasing number of studies have focused and contributed to the conceptualization (Millennium Ecosystem Assessment, Nature's contributions to people (NCP), CICES classification, ...) and the improvement of methodologies (see e.g., Manning et al., Garland et al., Hölting et al.) to assess ecosystem functions and services. Unfortunately, not much of these efforts were considered in the present study, which omit a proper classification and denomination of functions.

Except for crop yield (food production), none of the variable assessed represent really an ecosystem function, not talking about services (i.e., with a direct benefit for human), but are all some soil properties that might be used for soil quality assessment. None of them represent a process nor a function and more than half are relative abundances or diversity measures of soil microorganisms. The way the authors classified the assessed variables as proxies for ecosystem functions and group them in service categories ignore the large efforts made in the last decades in order to strengthened EMF studies and make them more comparable and relevant.

Although normalizing data for EMF analyses is standard, the results and interpretation of single functon(single variable) should not be done only relatively (e.g., % increase) but also on absolute level. Otherwise, no clear impact or implication can be drawn. Moreover, the way the data were scaled (0 to 1) results in small differences being "inflated" by this transformation. For example, what do an 11.6% increase in scaled C stock implies in reality, i.e., how much C was actually accumulated in the soil, and how much would it contribute to CO2 mitigation? By the way, no stocks were really assessed as only the

concentration of SOC was considered omitting bulk density measurement.

An important issue in EMF analyses is to avoid overrepresentation of specific functions (Manning et al.). In the present study, the “supporting functions” are dominating the analyses and might also argue that the variables soil carbon stocks, soil fertility, soil microbial habitat, soil biodiversity, soil decomposition and plant-soil symbiosis all contribute to the same ecosystem function/service of maintaining and regulating soil quality. If so, then these variables should be bundled and taken separately in the analyses.

Manning, P., et al. (2018). "Redefining ecosystem multifunctionality." *Nature Ecology and Evolution* 2(3): 427-436.

Hörling, L., et al. (2019). "Multifunctionality assessments – More than assessing multiple ecosystem functions and services? A quantitative literature review." *Ecological Indicators* 103: 226-235.

Garland, G., et al. (2020). "A closer look at the functions behind ecosystem multifunctionality: A review." *Journal of Ecology* 109(2): 600-613.

On cropping systems:

Conservation agriculture implies the three principles of minimal (zero) soil disturbance, permanent soil cover and diversified crop rotation to be applied. It is questionable if the experimental treatments with no tillage fulfill these requirements with exception of minimal soil disturbance. I mainly do not consider a wheat-maize crop rotation without additional elements such as cover crops or living mulches as diversified. It is also not clear from the management description, why the crop residues were exported in the tilled treatments. This might have substantially influenced the results in term of carbon input. The description of the management of the experimental systems is too scarce to understand the rationale behind it. For example, only the wheat management is described, what happened to the maize crop? Also, the amount of N input allocated to the wheat crop seems very high (285 kgN/ha) to ignore potential losses, which questions overall the sustainability of the systems on an environmental perspective (mostly if the absolute yield data are not available).

Interpretation:

Many of the conclusions are based on the fact that microbial biomass and SOC concentration increased by warming, and this slightly more under CA. However from the results and discussion, we see that there is a food production decline under CA with time, a reduced pathogen control potential under warming under CA, a higher soil fertility decrease with time under warming and CA. These are major negative trends that contradict the main claim of the study that CA would perform better under warming and decrease trade-offs between functions. Again, the fact that all results are presented as scaled data and compared on a relative basis, make any interpretation for concrete implementation difficult.

As example, climate mitigation is not properly assessed nor discussed as nothing is done nor discussed about the greenhouse gas (GHG) emissions from the different systems, which might be also high considering the amount of N input (N₂O emissions) and offset gains in SOC.

Overall, there is not much discussion on why and how CA would more benefit from warming than Tillage.

Although soil depth might be relevant for soil processes and plant growth, I do not see how to interpret this differentiation in term of ecosystem multifunctionality.

Detailed comments:

Given the main concerns raised in the general comments, I did not make a full detailed review of the manuscript.

Introduction

L58: There is probably not only one management measure to achieve climate-resilient agricultural production.

L89-96: I cannot completely follow and understand the rationale behind the main hypothesis that CA trigger more synergies and less trade-offs. Those synergies and trade-offs are not well stated in that section which describe some processes linked to warming effects.

Why would warming increase SOM decomposition more or only under CA? In what is a higher SOM decomposition beneficial and trigger more synergies (in the long-term)? SOM decomposition arguably sustain crop growth but would also mean higher potential losses, e.g., by greenhouse gas emissions.

L94, supporting the capability of microbes of doing what?

L102: Maize for food security? Feed product!

L104-106: The assessed “(agro-)ecosystem functions” are basically all soil derived properties or processes.

Results

L116: To which variable refers the P value? EMF index?

L119: These increases are all calculated on standardized values and are relative. No concrete conclusions can be done. Effective changes? Real impacts (meaning)?

L158: How can EMF influence individual functions over time, when it depends itself from the functions?

Method

How were multi-year data treated?

Split plot design in LMMs?

Figures

Figures are overloaded. Many times, one figure panel represent already a single figure.

Figure 2: Year 5 and 6 seem to have been special years in terms of crop yield and soil fertility, which strongly influence EMF and the warming effect. What were the reason for that? Also the symbiosis results are strongly influenced by the last assessment.

Figure 3: It seems CA does increase both, positive and negative links between functions.

Figure 5 might be used as graphical abstract but do not contain really data and from my point of view overemphasize the results of the study.

Reviewer #2

(Remarks to the Author)

Comments to the editor/author:

The manuscript entitled "Conservation agriculture supports ecosystem multifunctionality after decade-long warming" is a research article aiming to assess how artificial warming impacts soil multifunctionality across two agricultural management practices over a ten-year timespan at an experimental station in China. Overall, I think the research question is extremely timely and relevant for developing strategies to address the requirements of sustainable agriculture in our changing climate. Moreover, having an experimental set-up to explore these patterns under artificial warming for one decade is very rare and beneficial for better understanding these dynamics. That being said, however, I think there are some shortcomings to how the results from this experiment were assessed, and that need to be addressed prior to publication in Nature Communications.

First, there is very little to no discussion of what conservation agriculture entails, which I think is essential for understanding the implications of the study. Therefore, in the introduction I would give more information on what conservation agriculture actually is, and how different this is from what is traditionally practiced in this area of China. Considering how the treatments were described in the Methods section, it seems that you compared no-till to tilled systems. It is not clear how the conservation agriculture treatment is different from a no-till system, since the crop rotation and time with cover is the same. Moreover, I felt that the implications of your study was not properly discussed in the Discussion section. Why do you think this practice had such strong effects in your climate? How might this differ in other climates? You mention that warming actually improves many functions in your study, but this is likely not the same in arid regions. I would make this clear in the discussion how transferable these findings are to other global areas.

Secondly, you mention climate warming often, but what about other climatic variations associated with climate change (i.e. precipitation and wind patterns, etc.)? This should be mentioned at well. It is clear that you cannot adapt the experimental trial for this at this point, but I would include a section on the limitations of the study, and discuss how the interplay between heat and precipitation may differ in the future, and that this must also be taken into account when considering alternative management practices.

Additionally, I think the introduction and discussion sections needs to be restructured so that they are more straightforward and easier to follow. In the introduction, as mentioned above, the framework of conservation agriculture is not discussed, and thus the rationale for your hypothesis is not supported. There is also key information missing from your main aims section (L97-106): Where was the field experiment located? How many fields? How was the warming treatment implemented? Were the functions measured at one time point or multiple across the ten years? I think more information on the context of the experiment is needed for the reader to understand the larger framework of the project. Similarly, in the discussion section, the actual, practical significance of the results was unclear. In part this was related to how the sentences were worded (see details on this in the specific comments below), and in part this was due to how this section was structured. Rather than have extremely long, run-on paragraphs, I suggest breaking this section up into shorter sections, related to each function assessed and what the implications are for your study.

Finally, the English grammar needs to be re-checked. There were many errors throughout the text. Some specific examples are given below, but there are many others that still need to be fixed.

Specific Comments/Suggestions:

L43: How can you show that these functions are 'interdependent'?

L117: Is lower soil decomposition a good thing? In line 353 you describe how decomposition is a vital ecosystem service. Please resolve this contradiction.

L159: Food production in conservation agriculture decreased with time isn't this an alarming finding? Please discuss.

L684-691: Correlation is not causation. The fact that two variables are correlated does not necessarily mean they are linked.

L316-318: Could this also just mean that the functions are highly correlated? I would be careful with how you interpret these results.

L333-335: This sentence does not make sense. How can soil carbon stocks negatively effect context specificity and initial soil carbon content?

L334: How can soil carbon stocks 'compete' with crop yields for nutrients? Soil carbon stocks are a property.

L337-339: "However, we observed that warming decoupled the trade-offs and synergies between food production and other ecosystem functions and ensured stability of food production by protecting it from variations of other functions." What do you mean by 'protecting it from variations of other functions'? How does warming do that exactly?

L341-344: "In addition, food production was promoted by soil-borne plant pathogen control under ambient conditions, but the decoupling of food production and pathogen control under warming protected food production from reductions of pathogen control." The decoupling of these functions protected food production from reductions of pathogen control? I cannot follow what you mean here.

L44-47: I would re-word this sentence. It sounds like you are saying warming increased ecosystem multifunctionality under conservation practices. Is that what you mean? If so, please discuss this more in the discussion – what are the implications of this finding? Do you think it would be the same in other climatic zones?

L49: What are 'decoupled trade-offs'? Aren't trade-offs naturally coupled? If something is de-coupled there would be no benefit nor tradeoff.

L56: There are numerous effects and implications of climate change. I would therefore change 'consequence' to 'consequences'.

L60-61: Look at grammar in this sentence "...strategy.... have". I would change to "A sustainable management strategy described as conservation agriculture has been..."

L68: unknown

L93: Change 'include' to 'including'.

L100, L536: What does 'double-cropped' mean? Please explain. This implies two crops grown at once, rather than two crops in rotation, which I believe you are meaning.

L109, 572, 679: Here you use the term 'agroecosystem functions', and elsewhere you use 'ecosystem functions'. Either one is fine, but I suggest you use the same term throughout the text.

L116, 118: Change "Tables" to "Table".

L121: Please introduce what depth "topsoil" is considered in your study.

L131: remove "that".

L277: Change 'soil stocks' to 'soil C stocks'.

L157: The ecosystem functions were not 'subjected to warming', but rather the fields where the functions were measured. I would thus adapt the wording of this sentence.

L193, L229 and elsewhere: I would rephrase 'soil layers' to 'soil depths'. Layers indicates different soil horizons, whereas this is not necessarily the case in your particular experiment.

L319: Why do you use the term 'multi-functions' in this context? Isn't 'functions' the correct meaning in this case?

L548: Add 'is' between 'which' and 'predicted'.

L547-548: Can you confirm that the unheated plots did not receive some sort of heat effect from the heated treatments?

L572-574: Please specify what the abbreviations stand for in the text as well as the table.

L575-576: What do you mean by 'proportion of decomposer' and 'proportion of arbuscular mycorrhizal fungi'?

L577: What do you mean by 'opposite number of plant pathogen proportions'? Is this actually linked in any way to the

protection of the crops against pathogens? Do you have data from the crops grown showing rates of disease prevalence? If not, this value is very theoretical and not necessarily linked to pathogen protection.

L578: Change 'was' to 'were'.

L591: Do you mean 'and' instead of 'with'? Otherwise, what do you mean with 'respectively'?

L642: The proportion of soil-borne potential fungal plant pathogens compared to total fungi? Or total fungi + bacteria? It is not clear what it is a proportion of.

L650: Was this the min and max values for all samples together, or per treatment?

Tables and Figures:

While you provide a lot of key information, I think that the figures overall are too busy. I would reduce the number of graphs per figure to focus the reader on the most important findings. The extra graphs can be placed in the supplementary materials.

Supplementary:

Table 1: Change "Unite" to "Unit".

Table 1: What are the units for soil biodiversity?

Version 1:

Reviewer comments:

Reviewer #1

(Remarks to the Author)

I read carefully and with high expectations the revised manuscript considering the relevant topic and extensive experimental setup and data collection of the study but I have to admit that I was somehow disappointed about the few improvements done by the authors in relation to my previous review regarding the framework around ecosystem services and multifunctionality and the interpretation of the results.

Even if the authors mention that they carefully reflected the framework of the study in the revised version of the manuscript, I do not see noticeable changes in their understanding of the conceptualization of ecosystem services and multifunctionality, nor major changes in the analyses performed, their interpretations and practical significances (Responses A3, A5, A6, A10).

Although Tables S1 and S2 shows a classification of the assessed variables into international framework and terminology, the direct link and significance of these variables to reflect ecosystem services is still poor and not changed. The term ecosystem service is still directly used as equal to the assessed variables and is misleading. Services should be defined as the direct goods and benefits that human get from ecosystem, which according to table S2 would be at most 1) food production, 2) Climate regulation, 3) Habitat maintenance, 4) Maintenance of soil quality, and 5) disease regulation. Otherwise, it stays ecosystem functions or even properties.

Additionally:

Food production:

- Even if wheat yield is adequate to represent food production, I do not understand why maize data (second crop in the rotation) are not included.

Climate regulation:

- Even if soil C stock were now approximated using bulk density data (A9) and can be used as proxy for climate change mitigation, there is no information and interpretation of the real impact of the observed changes in relation to the potential compensation of emissions. Also, the assessed soil depths (in total 0-15 cm taken together) both represent top soil in my point of view. Now it is more and more supported that no tillage accumulates C at the surface but do not necessarily lead to increased C accumulation over the whole soil profile (Luo et al. 2010, Meurer et al. 2018, Ogle et al. 2019). There is now discussion about this in the manuscript.
- Additionally, I had concerns about the high N fertilization of the crops in relation to potential losses (gaseous, leaching) in the environment and overall sustainability of the cropping systems, which are not totally resolved by the author's answers (A14, A17). Theoretically, taking an average harvest index of 0.5 and nitrogen contents of 15% for grain and 7% for straw, we have an N uptake of less than 150 kg N /ha for the wheat crop. For an N input of 285 KgN/ha that means a surplus of 135 KgN prone to losses. Together with the 207 kg N/ha applied to maize, the production of over 400 kg mineral N applied yearly might also results in high energy use and GHG emissions.
- As demonstrated in the study, the increase in C stock might be more representative for and related to soil quality in general and microbial processes than climate mitigation.

Soil quality:

- The variable microbial biomass, soil fertility, AMF, and soil decomposition are all indicator for the same function and

service of maintaining/improving soil quality.

- However, I still miss the relevance of microbial biomass and relative abundances of ASVs to represent soil processes contributing to soil quality. For example, can really the small observed increase in AMF ASVs be linked to improved crop growth or better soil aggregation?
- The function soil fertility is defined as DOC and TN concentration. How these parameters relate to availability of nutrients for plant growth and an increase be really used as ecosystem service by e.g., reducing fertilization?

Disease control:

- It is difficult to interpret the changes in soil pathogen relative abundance and the real impact on natural disease control as ecosystem service. There is no information about the identity of the pathogens and their relevance as diseases for wheat or maize that could impact productivity.
- Additionally, it is not mentioned in the methods if plant protection products (e.g., fungicides) were used, which would make the contribution of natural soil disease suppression as ecosystem service very low and less relevant.

Finally, I do not follow the rationale and many times I simply do not understand the explanations (sentences) stated in the discussion. Here few examples:

L241: Warming decoupled the strong negative links between food production and soil biodiversity, plant-soil symbiosis and soil carbon stocks, and weakened the positive links between food production and soil-borne plant pathogen control under conservation agriculture.

L246: More synergies between soil C stocks and microbial attributes (including soil biodiversity, plant-soil symbiosis, and microbial habitat) were observed under conservation agriculture, and warming strengthened the cooperations ($P < 0.05$; Figs. 3a, 4b). Similar to topsoil, the cooperation between carbon stocks and soil biodiversity also existed in conservation agriculture in the subsoil.

What does cooperation mean? Is 5-15 really sub-soil?

L 257: The co-benefits between microbial attributes and trade-offs between pathogen control and microbial attributes occurred in both management practices in the two soil depths. Which co-benefits? Do you mean correlations? What does it means?

L 292: More importantly, warming-induced positive effects on yield strengthened increased with time. But still yield declined.

L 316: The improvement of food production and alleviation of its temporal tendency in conservation agriculture under warming actually benefit from 'climate-smart soil'. Alleviation of its temporal tendency? Is the experimental soil really "climate-smart"?

All these drawbacks behind the framework of the study and the little mechanistic/process explanations and discussions make it very difficult to derive real implications and a holistic impact assessment of the studied cropping systems in term of ecosystem services delivery.

Taking all these major concerns into account, I unfortunately cannot not support and recommend the manuscript for publication in Nature communications.

I still believe that there is a lot of interesting patterns behind the data that could deliver important insights on the effects of warming on agricultural soils. I recommend to focus more on temporal changes of the assessed variables and how this directly influence system performance in relation to the management of the systems (inputs).

other general comments:

The discission section reads more like a results section as it refers many times to figures. But do not really discuss the observations.

Many references to figures in the main text are not correct, e.g., L 342, 344, 379, 392.

Figure S4 is the same as Figure S8

The text would in general benefit language editing. To many sentences are difficult to follow or understand due to grammar and wording throughout the manuscript.

References:

Luo, Z.K., Wang, E.L., Sun, O.J., 2010. Can no-tillage stimulate carbon sequestration in agricultural soils? A meta-analysis of paired experiments. *Agric. Ecosyst. Environ.* 139, 224–231.

Meurer, K.H.E., Haddaway, N.R., Bolinder, M.A., Katterer, T., 2018. Tillage intensity affects total SOC stocks in boreo-temperate regions only in the topsoil-a systematic review using an ESM approach. *Earth Sci. Rev.* 177, 613–622.

Ogle, S.M., Alsaker, C., Baldock, J., Bernoux, M., Breidt, F.J., McConkey, B., Regina, K., Vazquez-Amabile, G.G., 2019.

Climate and soil characteristics determine where no-till management can store carbon in soils and mitigate greenhouse gas emissions. *Sci. Rep.* 9, 9.

Reviewer #2

(Remarks to the Author)

Thank you for this extremely thorough manuscript revision. I appreciate the care given to respond to the reviewer comments and suggestions, and think the updated version is greatly improved as a result of these changes. Aside from a few minor grammar/wording suggestions (see below), I think the manuscript is now ready for publication.

Minor suggestions:

L64: Add "and" before "is".

L72: I would change "uncertainty in climate projections" to "uncertainty in its efficacy under different future climate projections".

L83: Remove "growth".

L388, L142, and elsewhere: Remove the 's' from "managements".

L440: Add "although" in front of "climate".

L443: As this is a summary paragraph, I would replace the term "hydrothermal condition" with something more general and broadly associated with climate change, such as "precipitation patterns".

Version 2:

Reviewer comments:

Reviewer #2

(Remarks to the Author)

Thank you for this updated manuscript. Following the detailed revisions from the previous versions, I believe that the structure, approach and overall message and results are greatly improved.

The only change I would suggest is to add a few words at L182 to mention "how" soil health was calculated. This should not go into detail, but instead just give an indication of the general approach (i.e. Cornell Soil Health Assessment Scoring, or simply averaging the individual indicator scores).

Response to Reviewers' Comments

A. Response to Reviewer #1 (Remarks to the Author):

A1. The manuscript entitled “Conservation agriculture supports ecosystem multifunctionality after decade-long warming” presents an interesting study, which tackle two timely and relevant issues of agricultural production that are the holistic evaluation of different cropping systems on various ecosystem functions and their adaptation potential in face of climate change. In their study, the authors compare the impacts of conservation tillage (agriculture) and traditional cropping with tillage on several ecosystem properties over 10 years in a field experiment and how these impacts are affected by climate warming. Overall, the study and the field experiment are well designed and I acknowledge the huge efforts invested in conducting the warming experiment and the data collection. I also appreciate the analysis of synergies and trade-offs across the studied variables (even if I doubt about their real implications).

Response: We are grateful to the reviewer for the positive and supportive comments and for highlighting the strengths and novelty of our study.

A2. However, they are too many drawbacks and limitations in term of concept (EMF and cropping system classification), methodology (assessed variables as proxy for functions) as well as presentation and interpretation of the results in order to support this manuscript for publication (see below).

Response: We thank the reviewer for the suggestions to improve our manuscript. We have made substantial revisions to our manuscript after reflecting deeply on the reviewer's comments. We hope that the changes are appropriate and meet the standards for publication.

A3. I suggest the authors to rethink and narrow the framework of the study. Given the data availability, I would focus more on the effects of the experimental treatments on soil (microbiological) quality and how particular soil processes are impacted by warming.

Response: We thank the reviewer for their advice about the conceptual basis for our study which we have considered deeply. We have read very carefully the reviewer's recommended literature (A10) and used that to verify our classification of multiple services. We agree with the reviewer on the fundamental point that managing soil quality is the basis for delivering soil ecosystem services (Bünemann et al. 2018). A primary aim of our study is to illustrate the effect of 10 years of climate warming on the synergies and trade-offs between indicators of different soil ecosystem functions using a unique long-term dataset. Therefore, we have revisited our data in the context of the reviewer's comments and agree that a shift in focus to the insight that the parameters measured provide for specific soil ecosystem services and the relationships between them is possible. The reviewer will understand that our data is therefore inadequate for the direct evaluation of soil quality since this was not the conceptual basis for our experimental design, but we agree that our data does, either directly or indirectly, represent several critical ecosystem services provided by agricultural soil (Palm et al. 2014; Blaser et al. 2018; Tamburini et al. 2020; Guerra et al. 2022; Rillig et al. 2023), i.e., agricultural soil ecosystem services provided from cropland, beyond food production, are essential for the functioning of agroecosystems and for human wellbeing (Tamburini et al. 2020; Guerra et al. 2022). Therefore, based on reviewers' suggestions, we shift to focus on the effects of experiment treatment on soil ecosystem services and relationships among them, including those associated with soil microbiological quality (see Supplementary Table 1 for selected services). We refer the Reviewer to Supplementary Tables 1 and 2 (see below) which shows the relationship between ecosystem services from soil according to CICES and the variables that we measured. For

example, *Soil C Stocks* that are critical for carbon sequestration to mitigate climate warming as a component of soil organic matter (SOM; a key indicator for soil quality) is associated with *Soil Fertility* due to the nutrient pools that support *Food Production*. *Plant-Soil Symbiosis* and *Soil-borne Plant Pathogen Control* contribute to nutrient acquisition and maintenance of crop yield for *Food Production* (Delgado-Baquerizo et al. 2020; Martin & van der Heijden 2024; Pequeno et al. 2024). Again, we thank the reviewer for their advice which has helped us enrich our understanding of our dataset and hope that this compromise is acceptable to them.

The following is Supplementary Tables 1 and 2:

Supplementary Table 1 Ecosystem variables used to proxy soil ecosystem services in this study.

Soil ecosystem services	Variables	Unit	Importance
Food production	Crop yield	t ha ⁻¹	Indicating the delivery of food and economic benefits.
Soil C stocks	Soil C stocks	Mg C ha ⁻¹	Reflecting the soil carbon sequestration capacity of ecosystem in the long term and indicating the delivery of climate mitigation service ¹ .
Soil fertility	DOC	mg C kg ⁻¹	Build-up of nutrient pools that most frequently limit the growth of plants ² .
	TN	g N kg ⁻¹	
Soil microbial habitat	Microbial biomass carbon	ug C g ⁻¹	Reflecting the dynamics of soil quality and microbial activities ³ .
Soil biodiversity	Soil bacterial richness	unitless	Playing multiple roles in the delivery of ecosystem services, as a regulator of ecosystem processes, as a service in itself and as a good ³⁻⁷ .
	Soil fungal richness	unitless	
Plant-soil symbiosis	Proportion of potential Arbuscular mycorrhizal fungi (AMF)	%	Reflecting mycorrhizal colonization. Arbuscular mycorrhizal fungi establish associations with crops and provide many ecosystem services to agriculture ⁸⁻¹¹ . Beyond nutrient acquisition, AMF are helpful in drought tolerance, nutrient cycling, soil formation and aggregation, and therefore increase crop productivity ⁸⁻¹⁰ . Plant-soil symbiosis is important to overall agroecosystem multiservice via 'system performance and sustainability' that can reduce negative external inputs ⁹ .
Soil decomposition	Proportion of potential saprobes in fungal community	%	Reflecting soil decomposition capacity. Soil saprotrophic fungi fragment and decompose organic matter, making organically bound nutrients available for further processing through the entire soil food web and for plant uptake ¹⁰ .
Soil-borne plant pathogen control	Reciprocal of the proportion of soil-borne fungal potential plant pathogens	unitless	Reflecting soil health. Soils provide an array of potential reservoirs for fungal pathogen that damage food production ¹²⁻¹⁴ . Thus, soil-borne plant pathogen control can reflect the soil health contributing to food production ^{6,14} .

Supplementary Table 2 List of the studied soil ecosystem services and correspondence with Common International Classification of Ecosystem Services (CICES v5.2) ¹⁵.

Soil ecosystem services	CICES 5.2 group	CICES 5.2 section	NCP	MA	Goods and benefits
Food production	Cultivated terrestrial plants for nutrition, materials or energy.	Provisioning	Food and feed	Food	Harvested crop; Grain in farmer's store; flour, bread
Soil C stocks	Atmospheric composition and conditions.	Regulation & Maintenance	Regulation of air quality	Atmospheric regulation	Climate regulation resulting in avoided damage costs
Soil fertility	Regulation of soil quality.	Regulation & Maintenance	Formation, protection and decontamination of soils and sediments	Soil formation (supporting service)	Maintenance of soil quality and hence capability of soil for human use.
Microbial habitat	Regulation of soil quality.	Regulation & Maintenance	Formation, protection and decontamination of soils and sediments	Soil formation (supporting service)	Maintenance of soil quality and hence capability of soil for human use.
Soil biodiversity	Lifecycle maintenance, habitat and gene pool protection.	Regulation & Maintenance	Habitat creation and maintenance	No equivalent	Sustainable populations of useful or iconic species that contribute to a service in another ecosystem.
Plant-soil symbiosis	Regulation of soil quality.	Regulation & Maintenance	Formation, protection and decontamination of soils and sediments	Soil formation (supporting service)	Maintenance of soil quality and hence capability of soil for human use.
Soil decomposition	Regulation of soil quality.	Regulation & Maintenance	Formation, protection and decontamination of soils and sediments	Soil formation (supporting service)	Maintenance of soil quality
Soil-borne plant pathogen control	Pest and disease control.	Regulation & Maintenance	Regulation of organisms detrimental to humans	Disease regulation	Reduction in disease damage due to harvested fruit or vegetables.

NCP: nature's contributions to people. MA: Millennium Ecosystem Assessment.

References:

- Bender, S.F., Wagg, C., van der Heijden, M.G.A. (2016). An Underground Revolution: Biodiversity and Soil Ecological Engineering for Agricultural Sustainability. *Trends Ecol. Evol.* **31**, 440-452.
- Bünemann, E.K., Bongiorno, G., Bai, Z., Creamer, R.E., De Deyn, G., de Goede, R. *et al.* (2018). Soil quality – A critical review. *Soil Biol. Biochem.*, **120**, 105-125.
- Blaser, W.J., Oppong, J., Hart, S.P., Landolt, J., Yeboah, E. & Six, J. (2018). Climate-smart sustainable agriculture in low-to-intermediate shade agroforests. *Nat. Sustain.*, **1**, 234-239.
- Delgado-Baquerizo, M., Guerra, C.A., Cano-Díaz, C., Egidi, E., Wang, J.-T., Eisenhauer, N. *et al.*

(2020). The proportion of soil-borne pathogens increases with warming at the global scale. *Nat. Clim. Chang.*, 10, 550-554.

Guerra, C.A., Berdugo, M., Eldridge, D.J., Eisenhauer, N., Singh, B.K., Cui, H. *et al.* (2022). Global hotspots for soil nature conservation. *Nature*, 610, 693-698.

Martin, F.M. & van der Heijden, M.G.A. (2024). The mycorrhizal symbiosis: research frontiers in genomics, ecology, and agricultural application. *New Phytologist*, n/a.

Palm, C., Blanco-Canqui, H., DeClerck, F., Gatere, L. & Grace, P. (2014). Conservation agriculture and ecosystem services: An overview. *Agric. Ecosyst. Environ.*, 187, 87-105.

Pequeno, D.N.L., Ferreira, T.B., Fernandes, J.M.C., Singh, P.K., Pavan, W., Sonder, K. *et al.* (2024). Production vulnerability to wheat blast disease under climate change. *Nat. Clim. Chang.*, 14, 178-183.

Rillig, M.C., van der Heijden, M.G.A., Berdugo, M., Liu, Y.-R., Riedo, J., Sanz-Lazaro, C. *et al.* (2023). Increasing the number of stressors reduces soil ecosystem services worldwide. *Nat. Clim. Chang.*, 13, 478-483.

Tamburini, G., Bommarco, R., Wanger, T.C., Kremen, C., van der Heijden, M.G.A., Liebman, M. *et al.* (2020). Agricultural diversification promotes multiple ecosystem services without compromising yield. *Sci. Adv.*, 6, eaba1715.

A4. I also strongly suggest to not only present and discuss the data on a relative scale but also reflect actual absolute changes and their real implications for (soil) ecosystem functioning.

Response: We thank the reviewer for this important suggestion. We have added the absolute values of multiple soil ecosystem services in Supplementary Fig. 2 (see below) and modified the representation in main text based on reviewer's suggestions.

Supplementary Figure 2 Effects of warming and management on individual indicators of soil ecosystem services. (a) 0-5 cm soil depth; (b) 5-15 cm soil depth. Boxplots display the mean (horizontal line), the 25th and 75th percentiles (colored box), the minimum and maximum (whiskers). Conserv-Amb: conservation agriculture without warming; Conserv-Warm: conservation agriculture with warming; Conven-Amb: conventional agriculture without warming; Conven-Warm: conventional agriculture with warming.

We have also made the following changes to the Results and Discussion section:

Lines 192-195: “Overall, conservation agriculture had 1.62 times larger soil C stocks, 2.33 times greater soil fertility, 1.92 times increased microbial habitat, and 1.72 times better soil-borne plant pathogen control than conventional agriculture in topsoil under warming (Fig. 1; Supplementary Figs. 2 and 3).”

Lines 288-289: “Herein, we observed that warming increased crop yields by 9.3% and 11.2% under conservation agriculture and conventional agriculture, respectively”.

Lines 336-338: “Warming increased actual soil C stocks by 2.05% to 2.13% in the two soil depths under conservation agriculture (Supplementary Fig. 2).”

A5. Concept:

On ecosystem multifunctionality:

The importance of assessing the performance of agricultural production on various aspect of the ecosystem, other than productivity, is now broadly recognize and acknowledged. An important and increasing number of studies have focused and contributed to the conceptualization (Millennium Ecosystem Assessment, Nature’s contributions to people (NCP), CICES classification, ...) and the improvement of methodologies (see e.g., Manning et al., Garland et al., Hölting et al.) to assess ecosystem functions and services. Unfortunately, not much of these efforts were considered in the present study, which omit a proper classification and denomination of functions.

Response: We thank the reviewer for drawing our attention to the body of knowledge around EMF; these useful references are now cited in our manuscript. We agree that the evaluation of agricultural performance is now broadly focused on various aspects of the ecosystem, other now than productivity. We have read very carefully the references recommended by the Reviewer in the light of their advice and agree to refocus our study to ‘soil ecosystem services’. We have checked and verified the classification and denomination of the services used in our manuscript following CICES v5.2, Millennium Ecosystem Assessment (MA), and Nature’s contributions to people (NCP). Soil ecosystem services are defined from a human perspective and represent a wide range of benefits (including Soil C Stocks, Food Production, etc.) (Maestre et al. 2022; Rillig et al. 2023). We refer the Reviewer to the definitions in Supplementary Tables 1 and 2.

Food production: Food production is the main service of agroecosystem, indicating the delivery of food and economic benefits and belonging to provisioning service.

Soil C stocks: Soil C stocks reflect the soil carbon sequestration capacity of ecosystem in the long term, indicating the delivery of climate regulating service. Climate regulation makes human avoid damage costs in production and in daily life.

Soil decomposition: Soil decomposition reflects soil decomposition capacity for organic matter, making nutrients available for plant uptake (Guerra et al. 2022). This sub-service represents decomposition processes affecting soil quality, contributing to maintenance of soil quality and hence capability of soil for human use.

Soil fertility: Soil fertility is the build-up of nutrient pool that most frequently limit the growth of plants, which directly maintains soil quality and capability for human use and belongs to Regulation & Maintenance service.

Plant-soil symbiosis: Plant-soil symbiosis reflects mycorrhizal colonization of plants, contributing to nutrient acquisition, drought tolerance, nutrient cycling, soil formation and aggregation, and therefore increase crop productivity (Rillig et al. 2019; Guerra et al. 2022). Plant-soil symbiosis is important to overall agroecosystem multiservice via ‘system performance and sustainability’ that can reduce negative external inputs (Rillig et al. 2019). Plant-soil symbiosis contributes to maintenance of soil quality and hence capability of soil for human use. In CICES v5.2 classification, this sub-service is classed to maintenance of soil

structure by biological agents and belongs to group of regulation of soil quality.

Soil biodiversity: Soil biodiversity plays multiple roles in the delivery of ecosystem services, such as a regulator of ecosystem processes, as a service in itself and as a good (Mace et al. 2012; Zhou et al. 2022). In CICES v5.2 classification, this sub-service is classed to lifecycle maintenance, habitat and gene pool protection.

Microbial habitat: Microbial habitat reflects the dynamics of soil quality and microbial activities, contributing to maintenance of soil quality and hence capability of soil for human use.

Soil-borne plant pathogen control: Soil-borne plant pathogen control reflects soil health contributing to food production, which benefits reduction in disease damage to crops and is classed to pest and disease control (Delgado-Baquerizo et al. 2020; Pequeno et al. 2024).

In conclusion, we believe the proxies used in this manuscript is in proper classification and denomination, and can represent agricultural soil ecosystem service. We thank the reviewer again for providing these references, and we have cited these useful references in our manuscript.

References:

- Delgado-Baquerizo, M., Guerra, C.A., Cano-Díaz, C., Egidi, E., Wang, J.-T., Eisenhauer, N. *et al.* (2020). The proportion of soil-borne pathogens increases with warming at the global scale. *Nat. Clim. Chang.*, 10, 550-554.
- Guerra, C.A., Berdugo, M., Eldridge, D.J., Eisenhauer, N., Singh, B.K., Cui, H. *et al.* (2022). Global hotspots for soil nature conservation. *Nature*, 610, 693-698.
- Mace, G.M., Norris, K. & Fitter, A.H. (2012). Biodiversity and ecosystem services: a multilayered relationship. *Trends Ecol. Evol.*, 27, 19-26.
- Maestre, F.T., Le Bagousse-Pinguet, Y., Delgado-Baquerizo, M., Eldridge, D.J., Saiz, H., Berdugo, M. *et al.* (2022). Grazing and ecosystem service delivery in global drylands. *Science*, 378, 915-920.
- Pequeno, D.N.L., Ferreira, T.B., Fernandes, J.M.C., Singh, P.K., Pavan, W., Sonder, K. *et al.* (2024). Production vulnerability to wheat blast disease under climate change. *Nat. Clim. Chang.*, 14, 178-183.
- Rillig, M.C., van der Heijden, M.G.A., Berdugo, M., Liu, Y.-R., Riedo, J., Sanz-Lazaro, C. *et al.* (2023). Increasing the number of stressors reduces soil ecosystem services worldwide. *Nat. Clim. Chang.*, 13, 478-483.
- Rillig, M.C., Aguilar-Trigueros, C.A., Camenzind, T., Cavagnaro, T.R., Degrune, F., Hohmann, P. *et al.* (2019). Why farmers should manage the arbuscular mycorrhizal symbiosis. *New Phytologist*, 222, 1171-1175.
- Zhou, G., Lucas-Borja, M.E., Eisenhauer, N., Eldridge, D.J., Liu, S. & Delgado-Baquerizo, M. (2022). Understorey biodiversity supports multiple ecosystem services in mature Mediterranean forests. *Soil Biol. Biochem.*, 172, 108774.

A6. Except for crop yield (food production), none of the variable assessed represent really an ecosystem function, not talking about services (i.e., with a direct benefit for human), but are all some soil properties that might be used for soil quality assessment. None of them represent a process nor a function and more than half are relative abundances or diversity measures of soil microorganisms. The way the authors classified the assessed variables as proxies for ecosystem functions and group them in service categories ignore the large efforts made in the last decades in order to strengthened EMF studies and make them more comparable and relevant.

Response: We appreciate the reviewer's suggestions. We understand the reviewer concerns about limitations on ecosystem functions, which require biotic and abiotic processes. Therefore, we have now reshaped our framework toward a set of well acknowledge soil ecosystem services. We respectfully refer the reviewer to our previous responses to their comments: A3 and A5.

A7. Although normalizing data for EMF analyses is standard, the results and interpretation of

single function (single variable) should not be done only relatively (e.g., % increase) but also on absolute level. Otherwise, no clear impact or implication can be drawn.

Response: We thank the reviewer for this important suggestion. We have added the absolute values of multiple soil functions in Supplementary Fig. 2 and modified the representation in Result section. We respectfully refer the reviewer to our previous responses to their comments: A4.

A8. Moreover, the way the data were scaled (0 to 1) results in small differences being "inflated" by this transformation. For example, what do an 11.6% increase in scaled C stock implies in reality, i.e., how much C was actually accumulated in the soil, and how much would it contribute to CO₂ mitigation?

Response: We thank the reviewer for this important suggestion. We have added the absolute values of multiple soil functions in Supplementary Fig. 2 (see above) and modified the representation in Results and Discussion section:

Lines 192-195: "Overall, conservation agriculture had 1.62 times larger soil C stocks, 2.33 times greater soil fertility, 1.92 times increased microbial habitat, and 1.72 times better soil-borne plant pathogen control than conventional agriculture in topsoil under warming (Fig. 1; Supplementary Figs. 2 and 3)."

Lines 288-289: "Herein, we observed that warming increased crop yields by 9.3% and 11.2% under conservation agriculture and conventional agriculture, respectively".

Lines 336-338: "Warming increased actual soil C stocks by 2.05% to 2.13% in the two soil depths under conservation agriculture (Supplementary Fig. 2)."

A9. By the way, no stocks were really assessed as only the concentration of SOC was considered omitting bulk density measurement.

Response: Thanks for raising this basic point. It is regrettable that we didn't collect data of soil bulk density continuously (annually) to calculate soil C stocks strictly in this study. Bulk density is mainly a function of the parent material, in consideration of relatively constant of bulk density, we used mean value of bulk density (two sampling year) from Hou et al. (2018) to recalculate the soil carbon stocks. Consistently, we supplemented details in Method section and modified the tables and figures in manuscript and supplementary materials.

References:

Hou, R., Ouyang, Z., Han, D. & Wilson, G.V. (2018). Effects of field experimental warming on wheat root distribution under conventional tillage and no-tillage systems. *Ecology and Evolution*, 8, 2418-2427.

A10. An important issue in EMF analyses is to avoid overrepresentation of specific functions (Manning et al.). In the present study, the "supporting functions" are dominating the analyses and might could also argue that the variables soil carbon stocks, soil fertility, soil microbial habitat, soil biodiversity, soil decomposition and plant-soil symbiosis all contribute to the same ecosystem function/service of maintaining and regulating soil quality. If so, then these variables should be bundled and taken separately in the analyses.

Manning, P., et al. (2018). "Redefining ecosystem multifunctionality." *Nature Ecology and Evolution* 2(3): 427-436.

Hölting, L., et al. (2019). "Multifunctionality assessments – More than assessing multiple

ecosystem functions and services? A quantitative literature review." *Ecological Indicators* 103: 226-235.

Garland, G., et al. (2020). "A closer look at the functions behind ecosystem multifunctionality: A review." *Journal of Ecology* 109(2): 600-613.

Response: We thank the reviewer for drawing our attention to the body of knowledge around EMF; these useful references are now cited in our manuscript. We read these references very carefully and used the information to update our manuscript. We respectfully refer the reviewer to our previous responses (A3, A5) to their comments.

A11. On cropping systems:

Conservation agriculture implies the three principles of minimal (zero) soil disturbance, permanent soil cover and diversified crop rotation to be applied. It is questionable if the experimental treatments with no tillage fulfill these requirements with exception of minimal soil disturbance. I mainly do not consider a wheat-maize crop rotation without additional elements such as cover crops or living mulches as diversified.

Response: We apologize to the reviewer for unclear description of conservation agriculture. Conservation agriculture adopted in our study consists all three principles, including reduced or zero/no tillage, permanent soil cover by straw mulching, and wheat-maize crop rotation (Pittelkow et al., 2015; FAO, 2016; Kassam et al., 2019; Wittwer et al., 2021). The long-term experiment that we used was set up more than 14 years ago (2010) before the current research focus on diverse crop rotation as a tenet of conservation agriculture. We have added the definitions of the two management treatments to the Introduction section to provide clarity (Lines 62-63). We have also revised the Method section to make experimental treatment clearer.

Lines 63-65: "Conservation agricultural management encompasses reduced or zero/no tillage, permanent soil cover, and diverse crop rotations, is applicable in many different farming contexts^{2,10-12}".

Lines 704-709: "Winter wheat was seeded between 10 and 15 October, and harvested during the first 10 d of June. Then summer maize was seeded 5 d later and harvested during the first 10 d of October. After harvest, in the conventional agriculture treatment, the residues were removed and cultivation with a rotary tiller annually. In the conservation agriculture treatment, all residues were chopped to approximately 5 cm in length and retained on the soil surface, and adopted no tillage".

References:

FAO. (2016). *The State of Food and Agriculture: Climate Change, agriculture and food security*.

Kassam, A., Friedrich, T., Derpsch, R. (2019). Global spread of Conservation Agriculture. *J. Environ. Stud. Sci.* 76, 29-51.

Wittwer, R.A., et al. (2021). Organic and conservation agriculture promote ecosystem multifunctionality. *Sci. Adv.* 7, eabg6995.

Pittelkow, C.M., et al. (2015). Productivity limits and potentials of the principles of conservation agriculture. *Nature* 517, 365-368.

A12. It is also not clear from the management description, why the crop residues were exported in the tilled treatments. This might have substantially influenced the results in term of carbon input.

Response: We appreciate the reviewer's comments. We agree with the reviewer that surface residue retention can indeed contribute to soil organic matter and other benefits. Besides reduced or zero/no tillage, the main difference between conservation agriculture and

conventional agriculture (traditional tillage) is handling of crop residue. The conventional agriculture is based on local farming practice on the North China Plain which was to remove all crop residues at harvest for other purposes, i.e., heating and animal feed, and biofuel production. Conservation agriculture chopped crop residues returned to soil, and continuous soil cover by crop residues is one of the key characteristics of conservation agriculture. To avoid the misunderstanding, and also to make it clearer, we have replaced traditional tillage with conventional agriculture in the manuscript and figure legends throughout. We also added the detail information about these two management systems in our study, e.g.

Lines 112-114: “Herein, our study aimed to: (i) assess warming effects on multiple soil ecosystem services under conservation agriculture (permanent crop residues cover and no tillage) versus conventional agriculture (crop residue removal and annual tillage)”.

A13. The description of the management of the experimental systems is too scarce to understand the rationale behind it. For example, only the wheat management is described, what happened to the maize crop?

Response: We apologize to the reviewer that we did not provide the full details of the crop management to include the summer maize growing season. Full details of the experimental set up are provided in Method section. This has now been included.

Lines 704-721: “Winter wheat was seeded between 10 and 15 October and harvested during the first 10 d of June. Then summer maize was seeded 5 d later and harvested during the first 10 d of October. After harvest, in the conventional agriculture treatment, the residues were removed and the soil was cultivated with a rotary tiller annually. In the conservation agriculture treatment, all residues were chopped to approximately 5 cm in length and retained on the soil surface, and adopted no tillage. The conventional and conservation agriculture treatments had the same total N application rate, but part of the N application was from different sources. The total N application for conventional agriculture and conservation agriculture treatments in winter wheat growing seasons was $285 \text{ kg N ha}^{-1} \text{ yr}^{-1}$ following optimal application of fertilization^{67,68}. For conservation agriculture, the N input was 112.5 kg ha^{-1} of mineral fertilizer, 124.5 kg ha^{-1} of urea, and 48 kg ha^{-1} residue N. For conventional agriculture, the N input was 112.5 kg ha^{-1} of mineral fertilizer, and 172.5 kg ha^{-1} of urea. The total N application for conventional agriculture and conservation agriculture in summer maize growing seasons was $207 \text{ kg N ha}^{-1} \text{ yr}^{-1}$. For conservation agriculture, the N input was 175 kg ha^{-1} of urea and 32 kg ha^{-1} residue N. For conventional agriculture, the N input was 207 kg ha^{-1} of urea. All other management procedures were the same for conventional agriculture and conservation agriculture”.

A14. Also, the amount of N input allocated to the wheat crop seems very high (285 kgN/ha) to ignore potential losses, which questions overall the sustainability of the systems on an environmental perspective (mostly if the absolute yield data are not available).

Response: We appreciate the reviewer’s comments. The field experiment was set up to investigate the effect of management practices and warming, and all other management conditions according to local practice. We have added further details to the Method section about fertilizer management: “The total N fertilizer was $285 \text{ kg N ha}^{-1} \text{ yr}^{-1}$ in wheat growing season following optimal application of fertilization^{67,68}. The base fertilizer includes 112.5 kg ha^{-1} of N. Considering crop residue N input (48 kg ha^{-1} of N), for conservation agriculture, the inorganic N input was $124.5 \text{ kg ha}^{-1} \text{ yr}^{-1}$. For conventional agriculture, the remaining $172.5 \text{ kg ha}^{-1} \text{ yr}^{-1}$ of N was applied as urea.” (Lines 694-706). We also add the following contents of the sustainability of conservation agriculture on an environmental perspective in Discussion Section:

Lines 346-351: “However, some studies argued that adopting conservation agriculture may influence emissions of nitrous oxide, which with high global warming potential of CO₂ and can offset gains in soil C stocks^{13,53}. Short-term studies found that warming decreased N₂O emission in conservation agriculture fields, due to warming-induced higher uptake of NO₃⁻ and decreased denitrification rate⁵⁴. But long-term studies are need to verify and valid this phenomenon under warming conditions.”

A15. Interpretation:

Many of the conclusion are based on the fact that microbial biomass and SOC concentration increased by warming, and this slightly more under CA. However from the results and discussion, we see that there is a food production decline under CA with time, a reduced pathogen control potential under warming under CA, a higher soil fertility decrease with time under warming and CA. These are major negative trends that contradict the main claim of the study that CA would perform better under warming and decrease trade-offs between functions.

Response: We appreciate the reviewer’s comments. Although food production under conservation agriculture declined with time, warming alleviated the trend towards reduction (Supplementary Fig. 4). The warming-induced positive effects on food production increased with time (Fig. 2). By contrast, food production in conventional agriculture increased with time under ambient conditions, but the increasing trend was not significant under warming. In consideration of undifferentiated food production of the two systems both under ambient and warming conditions, we assumed that conservation agriculture performed better in the face of warming. Similarly, although soil fertility and pathogen control in conservation agriculture decreased under warming, they were higher in conservation agriculture than in conventional agriculture both under ambient and warming conditions during our observation. In conclusion, conservation agriculture has a better performance than conventional agriculture both in ambient and warming conditions. We have supplement it in Discussion Section (Lines 306-322).

A16. Again, the fact that all results are presented as scaled data and compared on a relative basis, make any interpretation for concrete implementation difficult.

Response: We thank the reviewer’s comments. We have added the absolute values of multiple soil services in supplementary materials and modified the representation in manuscript. We respectfully refer the reviewer to our previous responses (A7) to their comments.

A17. As example, climate mitigation is not properly assessed nor discussed as nothing is done nor discussed about the greenhouse gas (GHG) emissions from the different systems, which might be also high considering the amount of N input (N₂O emissions) and offset gains in SOC.

Response: We appreciate the reviewer’s comments. This study did not consider on greenhouse gas emissions, but this is an important focus of ongoing research at field experiment. Based on the data from the same field experiment, we found that warming increased total N uptake rates in both conventional agriculture and conservation agriculture. The enhanced N uptake could reduce N losses through increased uptake of NO₃⁻ and decreased denitrification rate under both conventional agriculture and conservation agriculture (Hou et al. 2018). Previous studies found that warming also decreased N₂O emission in conservation agriculture fields but increased it in conventional agriculture (Tu & Li 2017). This result could be due to crop straw cover instead of part of chemical N fertilizer reduced the inorganic N input and slowed decomposition of crop straw and thus decreased soil N₂O production. In addition, previous studies found lower CO₂ emission in conservation agriculture than conventional agriculture in wheat season both in ambient and warming conditions (Tu & Li 2017). There was no difference in CO₂ emission in maize season and annual cumulative CO₂ emission (Tu & Li 2017). There was no difference in CO₂ emission in maize season and annual cumulative CO₂ emission. On the whole, the

greenhouse gas emissions in conservation agriculture are basically flat, or even lower, than conventional agriculture both in ambient and warming conditions. These results verified the sustainability of conservation agriculture on an environmental perspective. On the whole, the greenhouse gas emissions in conservation agriculture are basically flat, or even lower, than conventional agriculture both in ambient and warming conditions. We have added relevant information about the trade-off between greenhouse gas (GHG) emissions and gains in SOC under in our discussion (Lines 346-351):

Lines 346-351: “However, some studies argued that adopting conservation agriculture may influence emissions of nitrous oxide, which with high global warming potential of CO₂ and can offset gains in soil C stocks^{13,53}. Short-term studies found that warming decreased N₂O emission in conservation agriculture fields, due to warming-induced higher uptake of NO₃⁻ and decreased denitrification rate⁵⁴. But long-term studies are need to verify and valid this phenomenon under warming conditions.”

References:

- Hou, R., Xu, X. & Ouyang, Z. (2018). Effect of experimental warming on nitrogen uptake by winter wheat under conventional tillage versus no-till systems. *Soil Tillage Res.*, 180, 116-125.
- Ju, X.T., Kou, C.L., Christie, P., Dou, Z.X. & Zhang, F.S. (2007). Changes in the soil environment from excessive application of fertilizers and manures to two contrasting intensive cropping systems on the North China Plain. *Environmental Pollution*, 145, 497-506.
- Tu, C. & Li, F. (2017). Responses of greenhouse gas fluxes to experimental warming in wheat season under conventional tillage and no-tillage fields. *Journal of Environmental Sciences*, 54, 314-327.

A18. Overall, there is not much discussion on why and how CA would more benefit from warming than Tillage.

Response: We thank the reviewer for this suggestion. We have added interpretations of the reason that conservation agriculture benefits more from warming than conventional agriculture in Introduction and Discussion sections. Because there are too many revisions, we cannot enumerate them and thanks for your patience of reading them throughout. Here are some examples:

Lines 83-89: “Warming that promotes plant growth above- and below-ground biomass²⁵ benefits soil fertility and carbon sequestration, enhancing microbial habitats and soil biodiversity, supporting further increases in plant growth²⁶. Crop residue retention in conservation agriculture promotes additional SOC formation by accelerating microbial turnover and necromass accumulation through organic matter supply, which also alleviates water and nutrient limitation for crops by improving soil quality^{17,22}.”

Lines 116-120: “under warming conditions, (i) conservation agriculture increases carbon input (including root and straw) and stimulates microbial growth, promoting organic carbon accumulation and nutrient availability. The improvement of soil C stocks and soil fertility raises soil biodiversity and pathogen control (because of pathogen dilution), consequently contributes to food production”.

Lines 293-306: “The productivity benefits of conservation agriculture may derive from favorable climate and soil conditions^{11,26,47}. Warming-induced extended growing season allow earlier planting, reduce cold injury and frost damage in seedlings, and later harvest^{42,47}, which directly improve crop growth and productivity⁴⁷. In addition, food production in conservation agriculture also benefits from enhanced organic matter decomposition, improved soil fertility, decouple trade-offs between food production and other services under warming (Figs. 1 and 3).

Warming increased higher plant carbon inputs (includes root biomass, exudation, and crop residues) to soil, and provide abundant substrates for microbial proliferation and activity under long-term conservation agriculture with warming⁴⁸. In addition, the reduced soil disturbance by conservation agriculture promotes aggregate formation and stability, providing a conducive environment for the preferential accumulation of microbial necromass and SOC formation and accrual^{16,17}. The improved soil quality thus increase food production and improve resilience to climate change¹.”

A19. Although soil depth might be relevant for soil processes and plant growth, I do not see how to interpret this differentiation in term of ecosystem multifunctionality.

Response: We appreciate the opportunity to calls for interpretation in a revised version of the manuscript. We have supplemented interpretations in Discussion section:

Lines 369-377: “Some studies argued that conservation agriculture is only beneficial to soil surface^{13,57}. Indeed, topsoil delivered higher soil services, especially soil C stocks, soil fertility, and microbial habitat than subsoil under conservation agriculture (Fig. 1; Supplementary Figs. 2 and 3). This may be due to differences in plant-derived C input in different soil depths. Besides root C input, topsoil is affected by crop residues from aboveground biomass¹⁷, stimulating higher soil microbial activity, soil C stocks and soil fertility directly. But our findings confirmed that conservation agriculture supported higher soil ecosystem multi-services than conventional agriculture regardless of soil depths (Fig. 1; Supplementary Fig. 3).”

A20. Detailed comments:

Given the main concerns raised in the general comments, I did not make a full detailed review of the manuscript.

Introduction

L58: There is probably not only one management measure to achieve climate-resilient agricultural production.

Response: We thank the reviewer for this suggestion. We have changed ‘a climate-resilient management measure’ to ‘climate-resilient land management measures’ in this sentence (Line 59).

A21. L89-96: I cannot completely follow and understand the rationale behind the main hypothesis that CA trigger more synergies and less trade-offs. Those synergies and trade-offs are not well stated in that section which describe some processes linked to warming effects.

Response: We thank the reviewer for advising on the improvement of hypothesis. Soil organic carbon and nutrient are the fundamental of soil ecosystem service. So, the improvement of soil nutrient and SOC under conservation agriculture may promote multiservice simultaneously and trigger synergies. The adequate soil substrate also prevents competition between various soil-based services delivering. We have reorganized and modified the hypothesis.

Lines 115-122: “We hypothesized that, under warming conditions, (i) conservation agriculture increases carbon input (including root and straw) and stimulates microbial growth, promoting organic carbon accumulation and nutrient availability. The improvement of soil C stocks and soil fertility raises soil biodiversity and pathogen control (because of pathogen dilution), consequently contributes to food production; (ii) improvement of soil C stocks and soil fertility induced by conservation agriculture triggers more synergies and less trade-offs; (iii) the benefits of conservation agriculture are cumulative over time.”

A22. Why would warming increase SOM decomposition more or only under CA? In what is a

higher SOM decomposition beneficial and trigger more synergies (in the long-term)? SOM decomposition arguably sustain crop growth but would also means higher potential losses, e.g., by greenhouse gas emissions.

Response: We thank the reviewer for this suggestion. We have improved the hypothesis in the main text, and this sentence was modified (Lines 115-122). We respectfully refer the reviewer to our previous responses (A17) to their comments about the trade-off of greenhouse emission and gains in SOC.

Lines 115-122: “We hypothesized that, under warming conditions, (i) conservation agriculture increases carbon input (including root and straw) and stimulates microbial growth, promoting organic carbon accumulation and nutrient availability. The improvement of soil C stocks and soil fertility raises soil biodiversity and pathogen control (because of pathogen dilution), consequently contributes to food production; (ii) improvement of soil C stocks and soil fertility induced by conservation agriculture triggers more synergies and less trade-offs; (iii) the benefits of conservation agriculture are cumulative over time.”

A23. L94, supporting the capability of microbes of doing what?

Response: We thank the reviewer for pointing out this lack of clarity. We have revised this sentence to make it clearer; it now reads:

Lines 116-118: “conservation agriculture increases carbon input (including root and straw) and stimulates microbial growth, promoting organic carbon accumulation and nutrient availability”.

A24. L102: Maize for food security? Feed product!

Response: We thank the review for pointing it out. We have reworded this sentence; it now reads:

Lines 126-127: “Our experiment was conducted using a typical crop rotation system (winter wheat (*Triticum aestivum* L.)-summer maize (*Zea mays* L.).)”

A25. L104-106: The assessed “(agro-)ecosystem functions” are basically all soil derived properties or processes.

Response: We thank the reviewer for making this important point. We respectfully refer the reviewer to our response A3 and A5 above. We have narrowed the framework and concentrated on soil ecosystem service.

A26. Results

L116: To which variable refers the P value? EMF index?

Response: We thank the review for pointing it out. The *P* value of mentioned variables and comprehensive index are all less than 0.05 (Figure 1; Supplementary Table 3).

A27. L119: These increases are all calculated on standardized values and are relative. No concrete conclusions can be done. Effective changes? Real impacts (meaning)?

Response: We thank the reviewer for making this important point. We have changed these valued to effective changes based on origin data.

Lines 173-175: “Overall, conservation agriculture had 1.62 times larger soil C stocks, 2.33 times greater soil fertility, 1.92 times increased microbial habitat, and 1.72 times better soil-borne plant pathogen control than conventional agriculture in topsoil under warming”.

A28. L158: How can EMF influence individual functions over time, when it depends itself from the functions?

Response: We apologize to the reviewer for the lack of clarity. We have revised this sentence to make it clearer; it now reads:

Lines 185-186: “Multiple soil ecosystem services varied substantially between years depending on managements and warming”.

A29. Method

How were multi-year data treated? Split plot design in LMMs?

Response: We apologize to the reviewer for the unclarity. Linear mixed models (LMMs) were used to assess the effects of warming and management practices on each soil ecosystem service and multiservice. In the LMMs, warming (0 for ambient temperature and 1 for warming) and management practices (0 for conservation agriculture and 1 for conventional agriculture) were considered as fixed effects, while sampling time (year) was termed as random intercept effects ($y \sim \text{warming} \times \text{management} + (1 | \text{year})$) (Wu et al., 2022) (Lines 837-844).

References:

Wu, L., *et al.* Reduction of microbial diversity in grassland soil is driven by long-term climate warming. *NAT. MICROBIOL* **7**, 1054-1062 (2022).

A30. Figures

Figures are overloaded. Many times, one figure panel represent already a single figure.

Response: We appreciate the reviewer’s comments. We have modified the figures and moved overload figures to supplementary materials. We hope the modification can satisfied review claims.

A31. Figure 2: Year 5 and 6 seem to have been special years in terms of crop yield and soil fertility, which strongly influence EMF and the warming effect. What were the reason for that? Also the symbiosis results are strongly influenced by the last assessment.

Response: We appreciate the reviewer’s comments. We look up climate data of the study area, and found that there were climatic anomaly of temperature and precipitation in the two years. Year 5 had a higher temperature than average annual temperature, and climatic conditions were favorable of crop yield (China Meteorology Administration 2015). While Year 6 was very abnormal accompanied with more frequent extreme weather and climate events influenced by super El Nino (China Meteorology Administration 2016). Conservation agriculture with warming had a good performance in maintenance of food production and soil fertility, mainly due to protection of permanent soil cover and no tillage. We have added some interpretation in Discussion section (Lines 313-320). The main function of plant-soil symbiosis is nutrient acquisition. So, we guess the soil fertility reduction in last year stimulated plant-soil symbiosis.

A32. Figure 3: It seems CA does increase both, positive and negative links between functions.

Response: We appreciate the reviewer’s comments. Because of recalculation of soil carbon

stocks, we recalculate the relationships between soil ecosystem services and modified Figure 3. The current result showed that conservation agriculture only increased positive links under warming.

A33. Figure 5 might be used as graphical abstract but do not contain really data and from my point of view overemphasize the results of the study.

Response: We appreciate the reviewer's comments. We have modified figure 5 and we still keep it as a conceptual model to make this manuscript readily intelligible for readers. We hope the reviewer will agree with us to do this.

Response to Reviewers' comments

B. Response to Reviewer #2 (Remarks to the Author):

B1. The manuscript entitled “Conservation agriculture supports ecosystem multifunctionality after decade-long warming” is a research article aiming to assess how artificial warming impacts soil multifunctionality across two agricultural management practices over a ten-year timespan at an experimental station in China. Overall, I think the research question is extremely timely and relevant for developing strategies to address the requirements of sustainable agriculture in our changing climate. Moreover, having an experimental set-up to explore these patterns under artificial warming for one decade is very rare and beneficial for better understanding these dynamics. That being said, however, I think there are some shortcomings to how the results from this experiment were assessed, and that need to be addressed prior to publication in Nature Communications.

Response: We appreciate the reviewer for supportive comments and for highlighting the strengths and the novelty of our study. We have made substantial revisions to our manuscript based on the reviewer's comments. We hope that the changes are appropriate and meet the standards for publication.

B2. First, there is very little to no discussion of what conservation agriculture entails, which I think is essential for understanding the implications of the study. Therefore, in the introduction I would give more information on what conservation agriculture actually is, and how different this is from what is traditionally practiced in this area of China.

Response: We appreciate the reviewer for the suggestions to emphasize the importance of conservation agriculture. We have added more information on conservation agriculture in Introduction section based on reviewer's comments.

Lines 63-69: “Conservation agricultural management encompasses reduced or zero/no tillage, permanent soil cover, and diverse crop rotations, is applicable in many different farming contexts^{2,10-12}. The demonstration of wide-ranging environmental benefits associated with its implementation, including improved soil quality that supports plant biomass production, increased soil organic carbon (SOC) stocks, and greater soil biodiversity^{2,7,8}, have led to adoption of conservation agriculture across 12.5% of arable land in one-third of countries worldwide¹⁰.”

B3. Considering how the treatments were described in the Methods section, it seems that you compared no-till to tilled systems. It is not clear how the conservation agriculture treatment is different from a no-till system, since the crop rotation and time with cover is the same.

Response: We thank the reviewer for raising this point and apologize for the lack of clarity. One of the aims of our study is to assess warming effects on multiple soil ecosystem services under conservation agriculture (permanent crop residues cover and no tillage) versus conventional agriculture (crop residue removed and annual tillage). To avoid the misunderstanding, and also to make it clearer, we have replaced traditional tillage with conventional agriculture in the manuscript and figure legends throughout the manuscript. We have also improved it in the method section (Lines 706-709).

Lines 706-709: “After harvest, in the conventional agriculture treatment, the residues were removed and the soil was cultivated with a rotary tiller annually. In the conservation agriculture treatment, all residues were chopped to approximately 5 cm in length and retained on the soil

surface, and adopted no tillage.”

B4. Moreover, I felt that the implications of your study was not properly discussed in the Discussion section. Why do you think this practice had such strong effects in your climate? How might this differ in other climates? You mention that warming actually improves many functions in your study, but this is likely not the same in arid regions. I would make this clear in the discussion how transferable these findings are to other global areas.

Response: We thank the reviewer for making this important point. We agree with the reviewer that the positive warming effects on ecosystems functions may context depended. We have added the limitations of our study according to the reviewer’s comment, as below:

Lines 440-447: “However, climate is an important driver of delivering of soil ecosystem services under conservation agriculture⁷, careful assessments of the regional climate are needed to determine the potential multi-benefits of adopting conservation agriculture^{7,12,13}. In view of hydrothermal condition, our findings may be generalized to other regions where water does not limit productivity (e.g., irrigated regions), then we propose that conservation agriculture can promote effective soil ecosystem multiservice under warming and increase agricultural resilience as a vital component of ‘climate-smart agriculture’.”

References:

- Sun, W., *et al.* Climate drives global soil carbon sequestration and crop yield changes under conservation agriculture. *Glob. Change Biol.* **26**, 3325-3335 (2020).
- Pittelkow, C.M., *et al.* Productivity limits and potentials of the principles of conservation agriculture. *Nature* **517**, 365-368 (2015).
- Powelson, D.S., *et al.* Limited potential of no-till agriculture for climate change mitigation. *Nat. Clim. Chang.* **4**, 678-683 (2014).

B5. Secondly, you mention climate warming often, but what about other climatic variations associated with climate change (i.e. precipitation and wind patterns, etc.)? This should be mentioned at well. It is clear that you cannot adapt the experimental trial for this at this point, but I would include a section on the limitations of the study, and discuss how the interplay between heat and precipitation may differ in the future, and that this must also be taken into account when considering alternative management practices.

Response: We thank the reviewer for making this important point. We agree with the reviewer that climate changes (i.e. precipitation and wind patterns) interacts with warming may lay important roles in regulating the observed effects here in conservation agriculture. We thank the reviewer for the excellent suggestions to improve the limitations of our study, as below:

Lines 440-447: “However, climate is an important driver of delivering of soil ecosystem services under conservation agriculture⁷, careful assessments of the regional climate are needed to determine the potential multi-benefits of adopting conservation agriculture^{7,12,13}. In view of hydrothermal condition, our findings may be generalized to other regions where water does not limit productivity (e.g., irrigated regions), then we propose that conservation agriculture can promote effective soil ecosystem multiservice under warming and increase agricultural resilience as a vital component of ‘climate-smart agriculture’.”

References:

- Sun, W., *et al.* Climate drives global soil carbon sequestration and crop yield changes under conservation agriculture. *Glob. Change Biol.* **26**, 3325-3335 (2020).
- Pittelkow, C.M., *et al.* Productivity limits and potentials of the principles of conservation agriculture. *Nature* **517**, 365-368 (2015).

Powlson, D.S., *et al.* Limited potential of no-till agriculture for climate change mitigation. *Nat. Clim. Chang.* **4**, 678-683 (2014).

B6. Additionally, I think the introduction and discussion sections needs to be restructured so that they are more straightforward and easier to follow. In the introduction, as mentioned above, the framework of conservation agriculture is not discussed, and thus the rationale for your hypothesis is not supported.

Response: We thank the reviewer for advising on the presentation of our manuscript. As advised, we have reviewed and made multiple changes in the Introduction section to provide a more integrated and cohesive narrative to give better context to our Introduction. There are too many changes to give all examples in the response letter, and we hope that the reviewer has time to read our Introduction section again and hope that the changes meet the reviewer's satisfaction. Here are some examples:

Lines 63-69: "Conservation agricultural management encompasses reduced or zero/no tillage, permanent soil cover, and diverse crop rotations, is applicable in many different farming contexts^{2,10-12}. The demonstration of wide-ranging environmental benefits associated with its implementation, including improved soil quality that supports plant biomass production, increased soil organic carbon (SOC) stocks, and greater soil biodiversity^{2,7,8}, have led to adoption of conservation agriculture across 12.5% of arable land in one-third of countries worldwide¹⁰."

Lines 83-89: "Warming that promotes plant growth above- and below-ground biomass²⁵ benefits soil fertility and carbon sequestration, enhancing microbial habitats and soil biodiversity, supporting further increases in plant growth²⁶. Crop residue retention in conservation agriculture promotes additional SOC formation by accelerating microbial turnover and necromass accumulation through organic matter supply, which also alleviates water and nutrient limitation for crops by improving soil quality^{17,22}."

B7. There is also key information missing from your main aims section (L97-106): Where was the field experiment located? How many fields? How was the warming treatment implemented? Were the functions measured at one time point or multiple across the ten years? I think more information on the context of the experiment is needed for the reader to understand the larger framework of the project.

Response: We thank the reviewer for this important recommendation. We have provided the detailed information for the field experiment and made the following revisions (Lines 141-152).

Lines 122-133: "To test this hypothesis, we conducted a decade-long field experiment to investigate the effects of soil management systems (conservation agriculture versus conventional agriculture) on multiple soil ecosystem services under experimental warming in North China Plain. Our experiment was conducted using a typical crop rotation system (winter wheat (*Triticum aestivum* L.)-summer maize (*Zea mays* L.)). Two levels of warming (ambient and +2 °C) were imposed using infrared heater according to soil warming predicted by IPCC greenhouse gas scenarios rates for northern China⁴². We simultaneously assessed eight soil ecosystem services annually to illustrate ecosystem multiservice and interactions across them, including food production, soil C stocks, soil fertility, microbial habitat, soil biodiversity, soil decomposition, plant-soil symbiosis, and soil-borne plant pathogen control (Supplementary Tables 1 and 2)".

B8. Similarly, in the discussion section, the actual, practical significance of the results was unclear. In part this was related to how the sentences were worded (see details on this in the

specific comments below), and in part this was due to how this section was structured. Rather than have extremely long, run-on paragraphs, I suggest breaking this section up into shorter sections, related to each function assessed and what the implications are for your study.

Response: We thank the reviewer for advising on the presentation of our manuscript. As advised, we have reviewed and made multiple changes in the Discussion section to provide a more integrated and clearer context. There are too many changes to give all examples in the response letter, and we hope that the reviewer has time to read our Discussion section again and hope that the changes meet the reviewer's satisfaction.

B9. Finally, the English grammar needs to be re-checked. There were many errors throughout the text. Some specific examples are given below, but there are many others that still need to be fixed.

Response: We thank the reviewer for their suggestions to improve our manuscript. We have checked the English grammar thoroughly and corrected all errors. Our native speaker co-authors have also checked the manuscript thoroughly to avoid the English grammar.

B10. Specific Comments/Suggestions:

L43: How can you show that these functions are 'interdependent'?

Response: We thank the reviewer for pointing this out. We adopted principal component analysis (PCA) (Supplementary Figure 11) and correlational analyses (Figure 3) to evaluate the correlations between these functions. In principal component analysis, the functions loaded on same direction of PC1 or PC2 axes mean interdependency among them. The correlational analyses were conducted to further clarify the synergies and trade-offs between functions, in which significant positive and negative correlations were recognized as synergies and trade-offs, respectively.

B11. L117: Is lower soil decomposition a good thing? In line 353 you describe how decomposition is a vital ecosystem service. Please resolve this contradiction.

Response: We apologize for the lack of clarity. For the whole agroecosystem, soil decomposition is a vital service, the higher the value and the more organic matter decomposing and higher nutrients available for further processing through the entire soil food web and for plant uptake (Bender et al., 2016) (Supplementary Table 1). However, in soil carbon sequestration processes, higher soil decomposition indicates more carbon decomposition and lower carbon accrual. In view of this, lower soil decomposition may be a good thing for soil carbon sequestration. We have moved the sentence describing soil decomposition into the paragraph illustrating soil C stocks changes.

Lines 159-161: "Warming promoted soil C stocks and microbial habitat under conservation agriculture (Figs. 1b, d, and 2a) but decreased the soil decomposition and soil-borne plant pathogen control in topsoil (Figs. 1h, l, and 2a)."

Reference:

Bender, S.F., Wagg, C., van der Heijden, M.G.A. An Underground Revolution: Biodiversity and Soil Ecological Engineering for Agricultural Sustainability. *Trends Ecol. Evol.* **31**, 440-452 (2016).

B12. L159: Food production in conservation agriculture decreased with time, isn't this an alarming finding? Please discuss.

Response: We appreciate the reviewer's comments. We have added relevant content to Discussion section based on reviewer's suggestions:

Lines 306-313: "Unfortunately, food production in conservation agriculture decreased with time, which may be due to reduction of soil-borne pathogen control (Supplementary Figure 5). Long-term warming appears to alleviate the tendency for reduced production with time under conservation agriculture, due to a higher increase in soil fertility and a decoupled trade-off with soil-borne plant pathogen control under warming (Supplementary Fig. 3). The improvement of food production and alleviation of its temporal tendency in conservation agriculture under warming actually benefit from 'climate-smart soil' ⁶."

B13. L684-691: Correlation is not causation. The fact that two variables are correlated does not necessarily mean they are linked.

Response: We appreciate the comment from the reviewer. We fully agree the reviewer that correlation is not causation, which may be only a statistical effect. Answering this question need to discuss the ecology implications behind the correlation. For example, we observed significant positive relationships between soil C stocks and soil biodiversity, microbial habitat, soil fertility and plant-soil symbiosis in conservation agriculture under warming (Figs. 3 and 4). This may be due to that, warming promotes plant growth and increases below ground biomass and thus more amount of plant input applied to soil (Qiu et al. 2021; Tian et al. 2024). The high plant inputs directly cause increase in soil fertility and carbon sequestration, as well as microbial habitat and biodiversity (Lines 85-89). Further, in view of fundamental role of soil C stocks in agroecosystem services, improvement of soil C stocks certainly promote soil biodiversity and microbial habitat. Another example is the negative relationship between soil C stocks and food production in conservation agriculture under ambient condition (Figs. 3 and 4). Although soil carbon sequestration is claimed to be a 'win-win' option to solve the challenges of climate warming and food security (Bradford et al. 2016; Qiao et al. 2022), there is no convincing empirical data to support general positive correlations between carbon sequestration and improvements in crop yield (Moinet et al. 2023). Further, soil C stocks can negatively affect crop yields, because of competition for nutrient between soil microbes and plants (van der Pol et al. 2022; Moinet et al. 2023), and context specificity (including climate and soil texture) (Moinet et al. 2023). Given this perspective, observed trade-offs between food production and C stocks might be anticipated (Lines 404-411). Collectively, correlation is not causation, but the ecology implications is essential for a deeper understanding of the mechanisms underlying the correlation, at least for the data we examined.

References:

- Bradford, M.A., Wieder, W.R., Bonan, G.B., Fierer, N., Raymond, P.A. & Crowther, T.W. (2016). Managing uncertainty in soil carbon feedbacks to climate change. *Nat. Clim. Chang.*, 6, 751-758.
- Moinet, G.Y.K., Hijbeek, R., van Vuuren, D.P. & Giller, K.E. (2023). Carbon for soils, not soils for carbon. *Glob. Change Biol.*, 1-15.
- Qiao, L., Wang, X., Smith, P., Fan, J., Lu, Y., Emmett, B. et al. (2022). Soil quality both increases crop production and improves resilience to climate change. *Nat. Clim. Chang.*, 12, 574-580.
- Qiu, Y., Guo, L., Xu, X., Zhang, L., Zhang, K., Chen, M. et al. (2021). Warming and elevated ozone induce tradeoffs between fine roots and mycorrhizal fungi and stimulate organic carbon decomposition. *Sci. Adv.*, 7, eabe9256.
- Tian, J., Dungait, J.A.J., Hou, R., Deng, Y., Hartley, I.P., Yang, Y. et al. (2024). Microbially mediated mechanisms underlie soil carbon accrual by conservation agriculture under decade-long warming. *Nat. Commun.*, 15, 377.
- van der Pol, L.K., Robertson, A., Schipanski, M., Calderon, F.J., Wallenstein, M.D. & Cotrufo, M.F. (2022). Addressing the soil carbon dilemma: Legumes in intensified rotations regenerate soil carbon while maintaining yields in semi-arid dryland wheat farms. *Agric. Ecosyst. Environ.*, 330, 107906.

B14. L316-318: Could this also just mean that the functions are highly correlated? I would be careful with how you interpret these results.

Response: We appreciate the comment from the reviewer. The degree, connectivity and evenness are all network properties. In data analysis, we converted the significant correlations ($P < 0.05$) between multiple services to a network graph object and analyzed the network properties. Networks of multiple-services relationships simultaneously account for all pairwise interactions and enables more intuitive in comparing the overall properties of synergies and trade-offs in different soil management systems (Felipe-Lucia et al. 2020). In terms of the same number of links, high evenness indicates that strength of different services has similar correlation strength, whereas low evenness indicates that strength of synergies/trade-offs is dominated by a few services. We carefully interpret this result in Discussion section.

Lines 391-393: “The higher degree, connectivity and evenness (Figs. 3c and d) suggested more balance between services were provided by conservation agriculture ⁶⁰.”

References:

Felipe-Lucia, M.R., Soliveres, S., Penone, C., Fischer, M., Ammer, C., Boch, S. et al. (2020). Land-use intensity alters networks between biodiversity, ecosystem functions, and services. *Proc. Natl. Acad. Sci. USA*, 117, 28140-28149.

B15. L333-335: This sentence does not make sense. How can soil carbon stocks negatively effect context specificity and initial soil carbon content?

Response: We apologize to the reviewer for the lack of clarity. We have reorganized this sentence and make it clear:

Lines 408-410: “Further, soil C stocks can negatively affect crop yields, because of competition for nutrient between soil microbes and plants ^{60,61}, and context specificity (including climate and soil texture) ⁶⁰.”

B16. L334: How can soil carbon stocks ‘compete’ with crop yields for nutrients? Soil carbon stocks are a property.

Response: We apologize to the reviewer for the lack of clarity. We have reorganized this sentence and make it clear:

Lines 408-410: “Further, soil C stocks can negatively affect crop yields, because of competition for nutrient between soil microbes and plants ^{61,62}, and context specificity (including climate and soil texture) ⁶¹.”

B17. L337-339: “However, we observed that warming decoupled the trade-offs and synergies between food production and other ecosystem functions and ensured stability of food production by protecting it from variations of other functions.” What do you mean by ‘protecting it from variations of other functions’? How does warming do that exactly?

Response: We appreciate the comment from the reviewer. In ambient conditions, these services were significantly related to food production, so variations of any services may lead to changes of food production under conservation agriculture. However, the services were unrelated to food production under warming, and thus changes of these services did not influence food production. Therefore, we assumed that ‘protecting it from variations of other services. We have improved this sentence to make it clearer:

Lines 411-413: “warming decoupled the trade-offs and synergies between food production and other ecosystem services and ensured stability of food production by protecting it from variations of other services.”

B18. L341-344: “In addition, food production was promoted by soil-borne plant pathogen control under ambient conditions, but the decoupling of food production and pathogen control under warming protected food production from reductions of pathogen control.” The decoupling of these functions protected food production from reductions of pathogen control? I cannot follow what you mean here.

Response: We apologize for the lack of clarity. According our results, food production is positively related to soil-borne plant pathogen control under ambient conditions. Based on our current knowledge, we can assume causal relations between food production and pathogen control (i.e. food production was promoted by soil-borne plant pathogen control). By contrast, food production is unrelated to soil-borne plant pathogen control under warming. Warming decreased soil-borne plant pathogen control in conservation agriculture. Therefore, the changes of soil-borne plant pathogen control did not influence food production due to the irrelevant relationship between them. That is, the decoupling of food production and pathogen control under warming protected food production from reductions of pathogen control. We have improved this sentence to make it clearer:

Lines 415-419: “In addition, we can assume causal relations between food production and pathogen control through positive relationship between them ⁴⁶. Thus, the decoupling of food production and pathogen control under warming protected food production from reductions of pathogen control.”

B19. L44-47: I would re-word this sentence. It sounds like you are saying warming increased ecosystem multifunctionality under conservation practices. Is that what you mean? If so, please discuss this more in the discussion – what are the implications of this finding? Do you think it would be the same in other climatic zones?

Response: We thank the reviewer for raising this point and apologize for the lack of clarity. We have added some contents in Discussion section based on Reviewer’s comments.

Lines 297-301: “This study provides empirical evidence for the advantages of conservation agriculture in maintaining ecosystem multiservice in a warming world, and recognizes potential of conservation agriculture in improving productivity, promoting adaptation to climate change and contributing to mitigation”.

Lines 440-447: “However, climate is an important driver of delivering of soil ecosystem services under conservation agriculture ⁷, careful assessments of the regional climate are needed to determine the potential multi-benefits of adopting conservation agriculture ^{7,12,13}. In view of hydrothermal condition, our findings may be generalized to other regions where water does not limit productivity (e.g., irrigated regions), then we propose that conservation agriculture can promote effective soil ecosystem multiservice under warming and increase agricultural resilience as a vital component of ‘climate-smart agriculture’.”

References:

- Sun, W., *et al.* (2020). Climate drives global soil carbon sequestration and crop yield changes under conservation agriculture. *Glob. Change Biol.* **26**, 3325-3335.
- Pittelkow, C.M., *et al.* (2015). Productivity limits and potentials of the principles of conservation agriculture. *Nature* **517**, 365-368.
- Powlson, D.S., *et al.* (2014). Limited potential of no-till agriculture for climate change mitigation.

Nat. Clim. Chang. **4**, 678-683.

B20. L49: What are ‘decoupled trade-offs’? Aren’t trade-offs naturally coupled? If something is de-coupled there would be no benefit nor tradeoff.

Response: We appreciate the comment from the reviewer. One aim of our study is to illustrate the effect of 10 years of climate warming on the synergies and trade-offs between indicators of different soil ecosystem functions using a unique long-term dataset. In this context, the ‘decoupled trade-offs’ under warming is relative to ambient conditions. We have improved this sentence to make it clearer:

Lines 47-50: “These benefits were related to warming-stimulated synergies between soil carbon stocks and microbial attributes, and warming-decoupled trade-offs between food production and soil carbon stocks and soil biodiversity.”

B21. L56: There are numerous effects and implications of climate change. I would therefore change ‘consequence’ to ‘consequences’.

Response: Sorry for this mistake. We have changed ‘consequence’ to ‘consequences’ (Line 55).

B22. L60-61: Look at grammar in this sentence “...strategy.... have”. I would change to “A sustainable management strategy described as conservation agriculture has been...”

Response: We appreciate the reviewer’s comment. We have changed this sentence followed the suggestions:

Lines 59-61: “A sustainable management strategy described as ‘conservation agriculture’ has been widely proposed as a nature-based solution to maintain food production and simultaneously promote multiple ecosystem services^{2,7-9}.”

B23. L68: unknown

Response: We appreciate the reviewer’s comment. We have changed ‘unkonw’ to ‘unknown’.

B24. L93: Change ‘include’ to ‘including’.

Response: We appreciate the reviewer’s comment. We have changed ‘include’ to ‘including’.

B25. L100, L536: What does ‘double-cropped’ mean? Please explain. This implies two crops grown at once, rather than two crops in rotation, which I believe you are meaning.

Response: We apologize to the reviewer for this imprecise expression, since we refer here to crop rotation. We have changed this throughout the manuscript.

B26. L109, 572, 679: Here you use the term ‘agroecosystem functions’, and elsewhere you use ‘ecosystem functions’. Either one is fine, but I suggest you use the same term throughout the text.

Response: We thank the reviewer for their advice. To avoid the misunderstanding, and followed comments of other reviewers and editor, we have used the term ‘soil ecosystem services’ throughout the text.

B27. L116, 118: Change “Tables” to “Table”.

Response: We appreciate the reviewer’s comment. We have changed “Tables” to “Table”.

B28. L121: Please introduce what depth “topsoil” is considered in your study.

Response: We appreciate the reviewer’s comment. We collected soil samples from 0-5 and 5-15 cm soil depths, respectively. So, 0-5 cm soil depth was considered as topsoil and 5-15 cm soil depth was considered as subsoil on our study. We have introduced topsoil in the main text when first use it:

Lines 144-145: “Conservation agriculture supported greater soil ecosystem services in topsoil (0-5 cm)”.

B29. L131: remove “that”.

Response: We appreciate the reviewer’s comment. We have removed “that” in this sentence.

B30. L277: Change ‘soil stocks’ to ‘soil C stocks’.

Response: We appreciate the reviewer’s comment. We have changed ‘soil stocks’ to ‘soil C stocks’.

B31. L157: The ecosystem functions were not ‘subjected to warming’, but rather the fields where the functions were measured. I would thus adapt the wording of this sentence.

Response: We appreciate the reviewer’s comment. We have re-wording this sentence.

Lines 185-186: “Multiple soil ecosystem services varied substantially between years depending on managements and warming”.

B32. L193, L229 and elsewhere: I would rephrase ‘soil layers’ to ‘soil depths’. Layers indicates different soil horizons, whereas this is not necessarily the case in your particular experiment.

Response: We appreciate the reviewer’s comment. We have changed ‘soil layers’ to ‘soil depths’ throughout the text.

B33. L319: Why do you use the term ‘multi-functions’ in this context? Isn’t ‘functions’ the correct meaning in this case?

Response: We apologize to the reviewer for this imprecise expression. Because of reframing of this study, we have changed the term ‘multi-functions’ to ‘services’.

B34. L548: Add ‘is’ between ‘which’ and ‘predicted’.

Response: We have added ‘is’ between ‘which’ and ‘predicted’.

B35. L547-548: Can you confirm that the unheated plots did not receive some sort of heat effect from the heated treatments?

Response: We appreciate the reviewer's comment. Yes, we can confirm that the unheated plots did not receive heat effects. There was a 5 m border between adjacent blocks and at least 10 m between the plots to avoid heating the control plots by the infrared radiators. We have filled the details in Method section.

Lines 701-703: "There was a 5 m border between adjacent blocks and at least 10 m between the plots to avoid heating the control plots by the infrared radiators."

B36. L572-574: Please specify what the abbreviations stand for in the text as well as the table.

Response: We appreciate the reviewer's comment. We have specified the abbreviations stand for in the text and supplementary table 1.

Lines 730-738: "We measured ten soil ecosystem variables to proxy eight soil ecosystem services, including food production (wheat yields), soil C stocks (soil carbon stocks), soil fertility (soil total nitrogen (TN) and dissolved organic carbon (DOC)), microbial habitat (microbial biomass carbon, MBC), soil biodiversity (richness of soil bacteria and fungi), soil decomposition (proportion of decomposer), plant-soil symbiosis (proportion of arbuscular mycorrhizal fungi), and soil-borne plant pathogen control (reciprocal of the proportion of soil-borne fungal potential plant pathogens) (Supplementary Tables 1 and 2)."

B37. L575-576: What do you mean by 'proportion of decomposer' and 'proportion of arbuscular mycorrhizal fungi'?

Response: We appreciate the reviewer's comment. Fungi typically live in highly diverse communities composed of multiple ecological guilds (also referred to as 'functional group'), such as saprotrophs (decomposer), arbuscular mycorrhizal fungi, and plant pathogens. FUNGuild is an open annotation tool for parsing fungal community datasets by ecological guild (Nguyen et al. 2016). We obtained the proportion of ecological guilds using FUNGuild database. The 'proportion of decomposer' and 'proportion of arbuscular mycorrhizal fungi' mean relative abundance of saprotrophs (decomposer) and arbuscular mycorrhizal fungi in fungal community.

Reference:

Nguyen, N.H., Song, Z., Bates, S.T., Branco, S., Tedersoo, L., Menke, J. et al. (2016). FUNGuild: An open annotation tool for parsing fungal community datasets by ecological guild. *Fungal Ecol.*, 20, 241-248.

B38. L577: What do you mean by 'opposite number of plant pathogen proportions'? Is this actually linked in any way to the protection of the crops against pathogens? Do you have data from the crops grown showing rates of disease prevalence? If not, this value is very theoretical and not necessarily linked to pathogen protection.

Response: We appreciate the reviewer's comment. The proportion of plant pathogen means relative abundance of plant pathogen in soil fungal community. We lacked data from the crops grown showing rates of disease prevalence, we selected this important variable based on previous reported studies (Delgado-Baquerizo et al. 2020; Pequeno et al. 2024). Previous studies have found that many of the most aggressive plant pathogens are soil-borne fungi that threaten food security as the chemical fungicides currently used against them are mostly ineffective (Delgado-Baquerizo et al. 2020; Pequeno et al. 2024). Opposite number of plant pathogen proportions means the inverse abundance of potential fungal plant pathogens, which was obtained by calculating the inverse of this variable (total relative abundance of fungal plant pathogens \times -1). Thus, the opposite number of plant pathogen proportions can represent plant pathogen control (Delgado-Baquerizo et al. 2020; Guerra et al. 2022).

References:

- Delgado-Baquerizo, M., Guerra, C.A., Cano-Díaz, C., Egidi, E., Wang, J.-T., Eisenhauer, N. *et al.* (2020). The proportion of soil-borne pathogens increases with warming at the global scale. *Nat. Clim. Chang.*, 10, 550-554.
- Guerra, C.A., Berdugo, M., Eldridge, D.J., Eisenhauer, N., Singh, B.K., Cui, H. *et al.* (2022). Global hotspots for soil nature conservation. *Nature*, 610, 693-698.
- Pequeno, D.N.L., Ferreira, T.B., Fernandes, J.M.C., Singh, P.K., Pavan, W., Sonder, K. *et al.* (2024). Production vulnerability to wheat blast disease under climate change. *Nat. Clim. Chang.*, 14, 178-183.

B39. L578: Change ‘was’ to ‘were’.

Response: We thank the Reviewer for pointing this out. We have changed ‘was’ to ‘were’.

B40. L591: Do you mean ‘and’ instead of ‘with’? Otherwise, what do you mean with ‘respectively’?

Response: We thank the Reviewer for pointing this out. We have deleted ‘respectively’ in the sentence.

B41. L642: The proportion of soil-borne potential fungal plant pathogens compared to total fungi? Or total fungi + bacteria? It is not clear what it is a proportion of.

Response: We appreciate the reviewer’s comment. We obtained the proportion of functional groups by parsing the fungal phylotypes using FUNGuild database, which specifically targets soil fungi. Thus, the proportion of soil-borne potential fungal plant pathogens compared to total fungi. We have re-wording this sentence to make it clear:

Lines 811-818: “We obtained the relative abundance of potential arbuscular mycorrhizal mutualists, saprotrophs, and plant pathotrophs from amplicon sequencing by parsing the soil fungal phylotypes using FUNGuild database⁸⁵. Fungi representing < 1% of the ASVs were not considered. Only highly probable and probable guilds were used in these analyses. Plant-soil symbiosis and soil decomposition were determined by the abundance of arbuscular mycorrhizal mutualists and saprotrophs. Soil-borne plant pathogen control was obtained by calculating reciprocal of the proportion of plant pathogen ($-1 \times$ the proportion of soil-borne potential fungal plant pathogens)^{76,86}.”

B42. L650: Was this the min and max values for all samples together, or per treatment?

Response: We appreciate the reviewer’s comment. To ensure multiple services and multiservice is comparable across treatments, this value is the minimum and maximum value for all samples. We have added details in Method sections (Lines 820-822).

B43. Tables and Figures:

While you provide a lot of key information, I think that the figures overall are too busy. I would reduce the number of graphs per figure to focus the reader on the most important findings. The extra graphs can be placed in the supplementary materials.

Response: We appreciate the reviewer’s comment. We have modified the figures and moved overload figures to supplementary materials based on reviewer’s comments.

B44. Supplementary:

Table 1: Change “Unite” to “Unit”.

Response: We appreciate the reviewer’s comment. We have changed ‘Unite’ to ‘Unit’ in supplementary table 1.

B45. Table 1: What are the units for soil biodiversity?

Response: We appreciate the reviewer’s comment. We used bacterial and fungal richness to represent soil biodiversity, and thus it was dimensionless.

Response to Reviewers' Comments

A. Reviewer #1 (Remarks to the Author):

A1. I read carefully and with high expectations the revised manuscript considering the relevant topic and extensive experimental setup and data collection of the study but I have to admit that I was somehow disappointed about the few improvements done by the authors in relation to my previous review regarding the framework around ecosystem services and multifunctionality and the interpretation of the results. Even if the authors mention that they carefully reflected the framework of the study in the revised version of the manuscript, I do not see noticeable changes in their understanding of the conceptualization of ecosystem services and multifunctionality, nor major changes in the analyses performed, their interpretations and practical significances (Responses A3, A5, A6, A10).

Response: We appreciate the Reviewer for supportive comments on the strengths and novelty of our study. We totally acknowledged the Reviewer's concerns about the framework around ecosystem services and multifunctionality, and regretted that the revised manuscript can't address these concerns. Given the data availability and Reviewer's two-round suggestions, we have made substantial revisions to our manuscript again. We shifted to focus on the insight that the soil parameters measured for soil quality instead of ecosystem services. While we agree with the reviewer's advice in general, we use the term 'soil health' rather than 'soil quality' because of the strong focus on biology in our study. The scope of soil health extends beyond human health to broader sustainability goals that include planetary health, which is broader than soil quality (Moebius-Clune et al., 2016; Lehmann et al., 2020). The main aims of this study were to investigate how management and warming affect soil health and crop yields mediated by soil microbial diversity and community composition.

We split the dataset (the previous first figure) into three parts to illustrate crop yields, soil health indicators, and microbial diversity (richness of bacteria, fungi, AMF, saproges, pathogens), respectively, respectively. To provide a valid soil health evaluation, we added soil data collected in 2017 (including 17 soil health indicators covered physics, chemistry and biology) and calculated a comprehensive soil health score to reflect the cumulative changes of soil health after long-term experimental warming. We also supplemented soil bacterial and fungal community composition data to get a full assessment of soil microbiomes. On this basis, we investigated the contribution of conservation agriculture compared with conventional agriculture to soil health indicators, microbial diversity, and crop production under long-term experimental warming. We found long-term warming increased soil health and crop productivity under conservation agriculture. Moreover, microbial biomass carbon and soil organic carbon benefited from conservation agriculture increased through time in face of climate warming. The increased soil health and shifts in soil fungal diversity consequently led to positive warming effects on crop yields over time under conservation agriculture. Our work demonstrates that conservation agriculture adapts better in soil health and crop yield in the face of a changing climate, which have important implications for sustainable resilient farming. We have tried our best to revise the manuscript according to the Reviewer's construction comments and suggestions, and made many changes in Introduction, Results, and Discussion. We hope that the Reviewer has time to read our manuscript again and the changes are appropriate and meet the Reviewer's satisfaction.

References

- Moebius-Clune, B.N., D.J. Moebius-Clune, B.K. Gugino, O.J. Idowu, R.R. Schindelbeck, A.J. Ristow, H.M. van Es, J.E. Thies, H.A. Shayler, M.B. McBride, K.S.M Kurtz, D.W. Wolfe, , Abawi, G.S. Comprehensive Assessment of Soil Health – The Cornell Framework, Edition 3.2. Cornell University (2016).
- Lehmann, J., Bossio, D.A., Kögel-Knabner, I., Rillig, M.C. The concept and future prospects of soil health. *Nat. Rev. Earth Environ.* 1, 544-553 (2020).

A2. Although Tables S1 and S2 shows a classification of the assessed variables into international framework and terminology, the direct link and significance of these variables to reflect ecosystem services is still poor and not changed. The term ecosystem service is still directly used as equal to the assessed variables and is misleading. Services should be defined as the direct goods and benefits that human get from ecosystem, which according to table S2 would be at most 1) food production, 2) Climate regulation, 3) Habitat maintenance, 4) Maintenance of soil quality, and 5) disease regulation. Otherwise, it stays ecosystem functions or even properties.

Response: We thank the Reviewer for this important suggestion. We have reorganized the data and manuscript based on soil health and crop yield changes under warming. We respectfully refer the Reviewer to our previous response A1.

A3. Additionally: Food production: Even if wheat yield is adequate to represent food production, I do not understand why maize data (second crop in the rotation) are not included.

Response: We thank the Reviewer for the suggestion. We have added maize data in Supplementary Fig. 1 (see below) and added representation in main text.

Supplementary Figure 1 Effects of warming and management on summer maize yield. **a** Average summer maize yield. Boxplots display the mean (horizontal line), the 25th and 75th percentiles (colored box), the minimum and maximum (whiskers). The data were analyzed based on five sampling years ($n = 15$ per treatment). Statistical analysis was performed using linear mixed model with sampling time as random factors. All reported P values result from two-sided statistical tests. Asterisk indicate significant differences in the warming effect of the individual management system as compared with their matched ambient condition. **b** Shift in the effect size of warming on crop yield over time for conservation and conventional agriculture, respectively. Linear regression model with two-sided test was used for the statistical analysis, and adjusted R-squared was used. Relationships are denoted with solid lines and fit statistics (R^2 and P values) for each management practice. The solid line represents the significant linear regression ($P < 0.05$), and the gray shading indicates the 95% confidence intervals. All reported P values result from two-sided statistical tests with * $P < 0.05$, ** $P < 0.01$, and *** $P < 0.001$. Conserv-Amb, conservation agriculture without warming; Conserv-Warm, conservation agriculture with warming; Conven-Amb, conventional agriculture without warming; Conven-Warm, conventional agriculture with warming.

Lines 165-169: “In contrast, the maize yield was only affected by soil managements ($P < 0.001$), while not influenced by warming (Supplementary Fig. 1; Supplementary Table 1). Conservation agriculture supported higher maize yield than conventional agriculture both under ambient and warming conditions ($P < 0.001$; Supplementary Fig. 1)”.

It is interesting to find that maize yield did not change response to experimental warming. We have supplemented interpretations in Discussion section:

Lines 346-349: “Warming had no effect on maize yields regardless of management type (Supplementary Fig. 1); the maize growing season is in summer (June to September) and C4 physiology is adapted to warmer temperatures and drier conditions¹³.”

A4. Climate regulation: Even if soil C stock were now approximated using bulk density data (A9) and can be used as proxy for climate change mitigation, there is no information and interpretation of the real impact of the observed changes in relation to the potential compensation of emissions.

Response: We fully agree the Reviewer's comments and admit the inadequate evaluation of climate regulation. We have acknowledged that SOC is preferable to reflect soil health and adaptation of agriculture to climate warming, not climate regulation or mitigation (Powlson *et al.* 2014). Now we have changed the focus into soil health evaluation and considered SOC as a soil health indicator. Hence, we have deleted information about climate regulation.

Reference:

Powlson, D.S., Stirling, C.M., Jat, M.L., Gerard, B.G., Palm, C.A., Sanchez, P.A. *et al.* (2014). Limited potential of no-till agriculture for climate change mitigation. *Nat. Clim. Chang.*, 4, 678-683.

A5. Also, the assessed soil depths (in total 0-15 cm taken together) both represent top soil in my point of view. Now it is more and more supported that no tillage accumulates C at the surface but do not necessarily lead to increased C accumulation over the whole soil profile (Luo *et al.* 2010, Meurer *et al.* 2018, Ogle *et al.* 2019). There is no discussion about this in the manuscript.

References:

Luo, Z.K., Wang, E.L., Sun, O.J., 2010. Can no-tillage stimulate carbon sequestration in agricultural soils? A meta-analysis of paired experiments. *Agric. Ecosyst. Environ.* 139, 224–231.

Meurer, K.H.E., Haddaway, N.R., Bolinder, M.A., Katterer, T., 2018. Tillage intensity affects total SOC stocks in boreo-temperate regions only in the topsoil—a systematic review using an ESM approach. *Earth Sci. Rev.* 177, 613–622.

Ogle, S.M., Alsaker, C., Baldock, J., Bernoux, M., Breidt, F.J., McConkey, B., Regina, K., Vazquez-Amabile, G.G., 2019. Climate and soil characteristics determine where no-till management can store carbon in soils and mitigate greenhouse gas emissions. *Sci. Rep.* 9, 9

Response: We agree with the Reviewer that the assessed 0-5 and 5-15 cm soil depths both represent topsoil and have deleted “topsoil” and “subsoil” in the manuscript. We have added some discussion about soil C accumulation over the soil profile in revised manuscript. We thank the Reviewer again for providing these references, and we have cited these useful references in our manuscript.

Lines 319-325: “Previous studies found that conservation agriculture accumulates soil C at the surface rather than the whole soil profile^{61,62}, or even changed distribution of C in the soil profile instead of increasing the total SOC^{14,63}. Yet our long-term monitoring only focuses on C accumulation at surface soil (0-15 cm) and may overestimate benefits in C sequestration over the whole soil profile. The changes of SOC in the deeper soil profiles need would be pay more attention in the future research.”.

A6. Additionally, I had concerns about the high N fertilization of the crops in relation to potential losses (gaseous, leaching) in the environment and overall sustainability of the cropping systems, which are not totally resolved by the author's answers (A14, A17). Theoretically, taking an average harvest index of 0.5 and nitrogen contents of 15% for grain and 7% for straw, we have an N uptake of less than 150 kg N /ha for the wheat crop. For an N input of 285 KgN/ha that means a surplus of 135 KgN prone to losses. Together with the 207 kg N/ha applied to maize, the production of over 400 kg mineral N applied yearly might also results in high energy use and GHG emissions.

Response: We appreciate the Reviewer's helpful comments. We have shifted to focus on soil health and considered SOC as a soil health indicator, rather than a surrogate of climate

regulation. We also deleted the sentences emphasized the climate mitigation by soil C stocks. This study did not consider on nitrogen losses and GHG emissions, but this is an important focus of ongoing research at field experiment.

A7. As demonstrated in the study, the increase in C stock might be more representative for and related to soil quality in general and microbial processes than climate mitigation.

Response: We appreciate the Reviewer for this useful suggestion. We have acknowledged that SOC is preferable to represent soil health, not climate regulation or mitigation (Powlson *et al.* 2014). We have reframed the study and considered SOC as a soil health indicator.

Reference:

Powlson, D.S., Stirling, C.M., Jat, M.L., Gerard, B.G., Palm, C.A., Sanchez, P.A. et al. (2014). Limited potential of no-till agriculture for climate change mitigation. *Nat. Clim. Chang.*, 4, 678-683.

A8. Soil quality: The variable microbial biomass, soil fertility, AMF, and soil decomposition are all indicator for the same function and service of maintaining/improving soil quality.

Response: We appreciate the Reviewer for this useful suggestion. We have reorganized this manuscript and respectfully refer the Reviewer to our previous response A1.

A9. However, I still miss the relevance of microbial biomass and relative abundances of ASVs to represent soil processes contributing to soil quality. For example, can really the small observed increase in AMF ASVs be linked to improved crop growth or better soil aggregation?

Response: We appreciate the Reviewer for this suggestion. Soil microbial biomass is the key determinant of nutrients availability that is crucial to plant growth and ecosystem functioning (Wardle 1998; Singh *et al.* 2016). Soil MBC and MBN are concluded to be frequently used as the key biological indicators of soil health that is sensitive to soil disturbances (Zuber & Villamil 2016; Bünemann *et al.* 2018). Soil fungi typically composed of multiple ecological guilds (also referred to as 'functional group'), such as saproges (decomposer), arbuscular mycorrhizal fungi (AMF), and plant pathogens. We obtained the richness and relative abundance of potential arbuscular mycorrhizal mutualists, saproges, and plant pathogens from amplicon sequencing by parsing the soil fungal phylotypes using FUNGuild database (Nguyen *et al.* 2016). Though it comes a range of pitfalls and potential biases, this tool is still one of the most recommended tools of assessing functional guilds and paves the way for ecologically informed analyses of fungal communities (Nilsson *et al.* 2019). There are many studies have provided evidences that fungal guilds are significantly and positively associated with multiple ecosystem functions, including nutrient cycling, decomposition, plant production, and reduced potential for pathogenicity (Delgado-Baquerizo *et al.* 2020; Wu *et al.* 2022).

We have reorganized this manuscript and supplemented the relationship between soil microbial diversity and community composition and soil health indicators (Fig. 4; Supplementary Fig. 9). In this study, we observed a decrease in saproge and total fungal richness in conservation agriculture under warming, which is correlated to soil health and crop yields. We found that the key indicators of soil health were tightly related to soil microbial richness and community composition (Figs. 4d-e; Supplementary Fig. 9), especially under conservation agriculture. Soil MBC and SOC were more relevant to soil fungal richness (especially saproge) and community composition compared with bacteria under conservation agriculture ($P < 0.05$). The intense relationship between soil fungi and carbon emphasized microbially mediated mechanisms underlie soil carbon accrual by conservation agriculture. Consequently, the increased MBC and decreased saproge richness jointly enhanced wheat yield under conservation agriculture with warming (lines 255-262, 271-277).

References:

- Wardle, D.A. (1998). Controls of temporal variability of the soil microbial biomass: A global-scale synthesis. *Soil Biol. Biochem.*, 30, 1627-1637.
- Singh, A., Singh, M.K. & Ghoshal, N. (2016). Microbial Biomass Dynamics in a Tropical Agroecosystem: Influence of Herbicide and Soil Amendments. *Pedosphere*, 26, 257-264.
- Delgado-Baquerizo, M., Reich, P.B., Trivedi, C., Eldridge, D.J., Abades, S., Alfaro, F.D. et al. (2020). Multiple elements of soil biodiversity drive ecosystem functions across biomes. *Nat. Ecol. Evol.*, 4, 210-220.
- Wu, L., Zhang, Y., Guo, X., Ning, D., Zhou, X., Feng, J. et al. (2022). Reduction of microbial diversity in grassland soil is driven by long-term climate warming. *NAT. MICROBIOL.*, 7, 1054-1062.
- Nilsson, R.H., Anslan, S., Bahram, M., Wurzbacher, C., Baldrian, P. & Tedersoo, L. (2019). Mycobiome diversity: high-throughput sequencing and identification of fungi. *Nat. Rev. Microbiol.*, 17, 95-109.
- Bünemann, E.K., Bongiorno, G., Bai, Z., Creamer, R.E., De Deyn, G., de Goede, R. et al. (2018). Soil quality – A critical review. *Soil Biol. Biochem.*, 120, 105-125.
- Zuber, S.M. & Villamil, M.B. (2016). Meta-analysis approach to assess effect of tillage on microbial biomass and enzyme activities. *Soil Biol. Biochem.*, 97, 176-187.
- Nguyen, N.H., Song, Z., Bates, S.T., Branco, S., Tedersoo, L., Menke, J. et al. (2016). FUNGuild: An open annotation tool for parsing fungal community datasets by ecological guild. *Fungal Ecol.*, 20, 241-248.

A10. The function soil fertility is defined as DOC and TN concentration. How these parameters relate to availability of nutrients for plant growth and an increase be really used as ecosystem service by e.g., reducing fertilization?

Response: We thank the Reviewer for making this important point. We fully acknowledge the deficiency in defining soil fertility as DOC and TN. To provide a valid soil health evaluation following the Review's comments, we added a full soil attributes dataset collected in 2017 to reflect the cumulative changes of soil health after long-term warming experimentation. In the full soil attributes dataset, soil DOC, DON, $\text{NH}_4^+\text{-N}$, $\text{NO}_3^-\text{-N}$, AP, and AK are all surrogates of nutrients availability for plant growth. We further calculated weights of individual soil indicators and recognized DOC, MBC, SOC, and TN as the key indicators illustrating soil health.

A11. Disease control: It is difficult to interpret the changes in soil pathogen relative abundance and the real impact on natural disease control as ecosystem service. There is no information about the identity of the pathogens and their relevance as diseases for wheat or maize that could impact productivity. Additionally, it is not mentioned in the methods if plant protection products (e.g., fungicides) were used, which would make the contribution of natural soil disease suppression as ecosystem service very low and less relevant.

Response: We thank the Review for pointing it out. We agree that it's inappropriate simply to define relative abundance of soil pathogen as a proxy of disease control service. We have considered pathogen richness as an influence factor of soil health and crop yield.

A12. Finally, I do not follow the rationale and many times I simply do not understand the explanations (sentences) stated in the discussion. Here few examples:

L241: Warming decoupled the strong negative links between food production and soil biodiversity, plant-soil symbiosis and soil carbon stocks, and weakened the positive links between food production and soil-borne plant pathogen control under conservation agriculture.

Response: We thank the Review for pointing it out. We have adjusted the focus of this study and rewrite the results. This sentence has been deleted.

A13. L246: More synergies between soil C stocks and microbial attributes (including soil

biodiversity, plant-soil symbiosis, and microbial habitat) were observed under conservation agriculture, and warming strengthened the cooperations ($P < 0.05$; Figs. 3a, 4b). Similar to topsoil, the cooperation between carbon stocks and soil biodiversity also existed in conservation agriculture in the subsoil.

What does cooperation mean? Is 5-15 really sub-soil?

Response: We thank the Review for pointing it out. We have shifted the focus of this study to soil health and crop yields, rather than soil ecosystem services and their relationships. We have rewritten the Results sections and this sentence has been deleted. We also deleted terms “subsoil”.

A14. L 257: The co-benefits between microbial attributes and trade-offs between pathogen control and microbial attributes occurred in both management practices in the two soil depths. Which co-benefits? Do you mean correlations? What does it means?

Response: We appreciate the Reviewer’s comments. We have adjusted the focus of this study and rewrite the results.

A15. L 292: More importantly, warming-induced positive effects on yield strengthened increased with time.

But still yield declined.

Response: We appreciate the Reviewer’s comments. This study aimed to compare differential responses of soil health and crop yield to warming under conservation and conventional agriculture, emphasizing the sustainability of conservation agriculture in a warmer world. Thus, we are more focused on tendency of warming-induced yield changes rather than changes of yield with time in the revised manuscript. We found that positive warming effects on wheat yield increased with time under conservation agriculture, but not conventional agriculture, emphasizing the cumulative benefits and sustainability of conservation agriculture in face of climate warming. We have deleted presentation and interpretation about time trend of crop yield, soil health, and soil microbial diversity in this revised version to avoid ambiguity in research focus.

A16. L 316: The improvement of food production and alleviation of its temporal tendency in conservation agriculture under warming actually benefit from ‘climate-smart soil’.

alleviation of its temporal tendency? Is the experimental soil really “climate-smart”?

Response: We appreciate the Reviewer’s comments. Our primary objective was to elucidate the differential response of soil health and crop yield to warming under conservation and conventional agriculture, emphasizing the sustainability of conservation agriculture in a warmer world. Thus, we are more focused on warming effects and tendency of warming-induced changes rather than changes of yield with time in the revised manuscript. We found that positive warming effects on wheat yield increased with time under conservation agriculture, but not conventional agriculture, emphasizing the cumulative benefits and sustainability of conservation agriculture in face of climate warming. Thus, we have deleted the content of temporal tendency of soil health indicator, crop yield and microbial diversity.

A17. All these drawbacks behind the framework of the study and the little mechanistic/process explanations and discussions make it very difficult to derive real implications and a holistic impact assessment of the studied cropping systems in term of ecosystem services delivery. Taking all these major concerns into account, I unfortunately cannot not support and recommend the manuscript for publication in Nature communications.

Response: We appreciate the Reviewer’s comments. We have tried our best to revise the manuscript according to the Review’s kind and construction comments and suggestions. We hope that the changes are appropriate and meet the standards for publication.

A18. I still believe that there is a lot of interesting patterns behind the data that could deliver important insights on the effects of warming on agricultural soils. I recommend to focus more on temporal changes of the assessed variables and how this directly influence system performance in relation to the management of the systems (inputs).

Response: We appreciate the Reviewer for supportive comments on the strength of this dataset and novelty of our study. We have reframed this study according to the Reviewer's comments. Please see response A1.

A19. other general comments:

The discussion section reads more like a results section as it refers many times to figures. But do not really discuss the observations.

Response: We apologize to the Reviewer for insufficiently synthetic discussion about the observations. We have reframed this study and rewrite Discussion according to the Reviewer's comments. There are too many changes and we hope that the Reviewer has time to read our Discussion section again. We hope that the changes are appropriate and meet the standards for publication.

A20. Many references to figures in the main text are not correct, e.g., L 342, 344, 379, 392.

Response: We apologize to the Reviewer for careless. We have checked the references to figures thoroughly the main text to avoid incorrect references.

A21. Figure S4 is the same as Figure S8

Response: We apologize to the Reviewer for careless. We have checked the Supplementary figures thoroughly to avoid these mistakes.

A22. The text would in general benefit language editing. To many sentences are difficult to follow or understand due to grammar and wording throughout the manuscript.

Response: We appreciate the Reviewer's comments. This manuscript has been revised extensively according to the Reviewer's constructive suggestions. In addition, the expression of the manuscript has been improved with the help of native English speakers.

Response to Reviewers' comments

B. Reviewer #2 (Remarks to the Author):

B1. Thank you for this extremely thorough manuscript revision. I appreciate the care given to respond to the Reviewer comments and suggestions, and think the updated version is greatly improved as a result of these changes. Aside from a few minor grammar/wording suggestions (see below), I think the manuscript is now ready for publication.

Response: We sincerely thank the Reviewer for thoroughly examining our manuscript and supportive comments.

B2. Minor suggestions: L64: Add “and” before “is”.

Response: We thank the Reviewer for pointing this out. We have added “and” before “is” (line 65).

B3. L72: I would change “uncertainty in climate projections” to “uncertainty in its efficacy under different future climate projections”.

Response: We appreciate the Reviewer's suggestion. We have changed “uncertainty in climate projections” to “uncertainty in its efficacy under different future climate projections”. This sentence now reads:

Lines 69-72: “However, a paucity of systematic and quantitative assessments of the impacts of long-term climate warming on the potential of conservation agriculture to support soil health and crop production creates uncertainty in its efficacy under different future climate projections^{13,14}.”

B4. L83: Remove “growth”.

Response: We thank the Reviewer for pointing this out. Because of adjustment of the content, this sentence has been deleted.

B5. L388, L142, and elsewhere: Remove the ‘s’ from “managements”.

Response: We appreciate the Reviewer's suggestion. We have changed “managements” into “management” and checked the manuscript thoroughly.

B6. L440: Add “although” in front of “climate”.

Response: We appreciate the Reviewer's suggestion. We have reorganized this sentence to make the expression concise.

Lines 379-382: “However, we recognize that local climatic conditions are an important driver of soil health and crop yield⁷ and careful regional assessments are needed when considering the potential consequences of adopting conservation agriculture^{7,13}.”

B7. L443: As this is a summary paragraph, I would replace the term “hydrothermal condition” with something more general and broadly associated with climate change, such as “precipitation patterns”.

Response: We thank the Reviewer for this important suggestion. Because of adjustment of the content, this sentence has been deleted.

Response to Reviewers' Comments

Reviewer #2 (Remarks to the Author):

Thank you for this updated manuscript. Following the detailed revisions from the previous versions, I believe that the structure, approach and overall message and results are greatly improved.

Response: We sincerely thank the Reviewer for thoroughly examining our manuscript and supportive comments.

The only change I would suggest is to add a few words at L182 to mention "how" soil health was calculated. This should not go into detail, but instead just give an indication of the general approach (i.e. Cornell Soil Health Assessment Scoring, or simply averaging the individual indicator scores).

Response: We appreciate the Reviewer for this useful suggestion. We have added more information about calculation of soil health. This sentence now reads:

Lines 181-183: The soil health score (Cornell Soil Health Assessment Scoring) was affected by management, warming, and their interactions ($P < 0.05$; Supplementary Table 2).